

# Modeling the Greenland englacial stratigraphy

Andreas Born[1,2] and Alexander Robinson[3,4,5]

[1]Department of Earth Science, University of Bergen, Norway
[2]Bjerknes Centre for Climate Research, Bergen, Norway
[3]Complutense University of Madrid, Spain
[4]Geosciences Institute, CSIC-UCM, Madrid, Spain
[5]Potsdam Institute for Climate Impact Research, Potsdam, Germany

**Correspondence:** Andreas Born (andreas.born@uib.no)

**Abstract.** Radar reflections from the interior of the Greenland ice sheet contain a comprehensive archive of past accumulation rates and ice dynamics. Combining these data with dynamic ice sheet models may greatly aid model calibration, improve past and future sea level estimates, and enable insights into past ice sheet dynamics that neither models nor data could achieve alone. Unfortunately, simulating the continental-scale ice sheet stratigraphy represents a major challenge for current ice sheet
models. In this study, we present the first three-dimensional ice sheet model that explicitly simulates the Greenland englacial stratigraphy. Individual layers of accumulation are represented on a grid whose vertical axis is time so that they do not exchange mass with each other as the flow of ice deforms them. This isochronal advection scheme is independent from the ice dynamics and only requires modest input data from a host thermomechanical ice-sheet model, making it easy to transfer to a range of models. Using an ensemble of simulations, we show that direct comparison with the dated radiostratigraphy data yields notably
more accurate results than selecting simulations based on total ice thickness. We show that the isochronal scheme produces a more reliable simulation of the englacial age profile than traditional age tracers. The interpretation of ice dynamics at different times is possible but limited by uncertainties in the upper and lower boundaries conditions, namely temporal variations in surface mass balance and basal friction.

## 1 Introduction

Summarizing the history of oceanography, Wunsch (2007) describes how the complicated retrieval of observations in the pioneering period caused the field to take on a 'geological flavor'. The necessity to combine data from expeditions spanning various decades painted the picture of a largely invariable ocean with constant strata of water mass properties. Later, advances in measurement techniques, most notably the advent of space-borne remote sensing, forced a revision of this view and it is
now widely accepted that the ocean is capable of even abrupt dynamic changes in which the vertical stratification often plays a pivotal role (e.g., Li and Born, 2019). The proverbially glacial flow of land ice may be less prone to changing quite as rapidly as the ocean, but the scarcity of data from the ice sheet's interior until recently mirrored that of early oceanography. In this





case, air-borne remote sensing provided a first comprehensive dataset using radar soundings (MacGregor et al., 2015) and it was shown that the structure of this *radiostratigraphy* also contains information about ice dynamics and how it changed over time (MacGregor et al., 2016).

Modeling techniques must accompany this progress. The analysis of dynamic signals in the new radiostratigraphy dataset relied on a steady-state model of ice dynamics published 40 years prior (Whillans, 1976), predating the development of all thermomechanical ice sheet models. The spatially comprehensive simulation of englacial layers could aid the site selection for ice cores to be drilled in dynamically active regions (EGRIP), where to find intact stratigraphy near the glacier bed (Fischer et al., 2013), to reconstruct past accumulation rates by decomposing dynamic thinning from the depositional signal (Wadding-ton et al., 2007), or to determine the origin and age of outcrops near the margins (Higgins et al., 2015). Not least would models greatly benefit from calibration and validation with the new wealth of observational constraints.

Previous efforts to simulate the englacial layering and age profiles in three-dimensional ice sheet models include Eulerian age tracers (Greve, 1997), Lagrangian particle tracking (Rybak and Huybrechts, 2003), and semi-Lagrangian tracer advection schemes (Tarasov and Peltier, 2003). The accuracy of Eulerian tracer advection is severely reduced by spurious diffusion in finite-differences numerical schemes and fully Lagrangian methods suffer from low particle densities in the most interesting region near the glacier bed. Semi-Lagrangian schemes have seen some success in simulating the three-dimensional stratigraphy of the Greenland ice sheet (GrIS) (Clarke et al., 2005; Lhomme et al., 2005; Martín et al., 2009; Goelles et al., 2014). More recent work proposed a fourth approach by simulating layers of equal age, isochrones, explicitly (Born, 2017). Here, the vertical grid is no longer defined in space as, e.g. on terrain-following coordinates, but in time, representing individual periods of accumulation. Over time, isochronal layers will thin as ice flows toward the margins which allows younger layers to subside, but no flow crosses the vertical boundaries of the grid.

Here we generalize the isochronal modeling approach by addressing its shortcoming and applying it to the entire GrIS. In the formulation by Born (2017), the isochronal model is a stand-alone ice sheet model that calculates all of its properties on the very finely layered isochronal grid. This greatly increases the computational cost with only marginal benefit because most variables such as temperature and velocity do not vary over such small vertical scales. As a consequence, the original isochronal model only covered a two-dimensional section through the GrIS. In this study we will decouple the calculation of the isochrone thickness from the model physics. Information on ice flow is now obtained from the comprehensive thermomechanical model Yelmo (Robinson et al., 2020). This strategy resembles that of Clarke et al. (2005), where the isochrone model becomes a tracer transport scheme coupled to a conventional model of ice dynamics. One of the major advantages is that the new modularity allows for easy coupling with other ice sheet models.

After a detailed description of the model and experimental setup in section 2, we will show how the simulation of isochrones can be used to constrain an ensemble of transient simulations of the last 160,000 years (160 kyr) in section 3. Individual model parameters, in particular those controlling accumulation at different times during the simulation create spatially complex patterns that also differ on the various isochronal surfaces. These additional constraints result in a simulation that is notably different from one that only considers the present day ice thickness as a tuning target. Comparison of the isochronal tracer





scheme with a second-order Eulerian age tracer scheme highlights the shortcomings of the latter. We discuss our results and conclude in section 4 and outline future applications.

## 2 Methods

### 2.1 The ice sheet model Yelmo

Yelmo is a hybrid ice-sheet model, which heuristically sums the shallow-ice and shallow-shelf approximations (SIA and SSA, respectively) to obtain the ice velocity at a given location (Bueler and Brown, 2009). It is thermomechanically coupled, and it employs Glen's flow law with an exponent of n=3. The model has been run at a horizontal resolution of 32 km with 20 vertical layers. Additional details of the model can be found elsewhere (Robinson et al., 2020), however here we will describe the key model properties that are relevant to this study.

#### 2.1.1 Basal friction

Basal friction at the ice-bed interface is represented with a linear friction law,

$$\tau_b = \beta u_b, \tag{1}$$

with basal velocity $u_b$ and coefficient

$$\beta = \frac{c_b}{u_0} N. \tag{2}$$

Here, $c_b$ is a unitless 2D field representing the basal properties under the ice, $u_0 = 100\mathrm{m/a}$ is a scaling constant and $N = \rho g H$ is the effective pressure of the ice. $N$ thus evolves with the ice sheet, modifying friction over time, while $c_b$ and $u_0$ remain fixed. As will be described below $c_b$ is optimized to reduce the mismatch of modeled and observed ice thickness at the present day.

#### 2.1.2 Enhancement factor

Empirical flow laws of ice sheet models are commonly modified with an enhancement factor, $E$, reflecting changing ice softness depending on flow regime and background climatic conditions (Paterson, 1991; Ma et al., 2010). $E$ evolves in space and time. Slow flow of grounded ice driven by vertical shearing, mainly near the domes, is enhanced due to single-maximum direction fabric growth. In ice streams and ice shelves, preferred directions in the ice fabric are removed, making the ice stiffer (Ma et al., 2010). Glacial-age ice in vertical-shear regimes has also been observed to be softer than interglacial-age ice, due to a higher concentration of impurities that hinder crystal growth, which further enhances flow in these regions (Paterson, 1991).



To simulate this in a straightforward way, we consider the enhancement factor as a 3D field defined by two contributions: the reference enhancement factor as defined by the flow regime ($E_{\mathrm{ref}}$) and additional enhancement resulting from the evolution of the ice sheet in time ($E_t$):

$$E = E_{\mathrm{ref}} E_t. \tag{3}$$

To calculate $E_{\mathrm{ref}}$, we use three parameters, which define the enhancement factor for purely shearing, streaming and floating regimes ($E_{\mathrm{shr}}$, $E_{\mathrm{strm}}$ and $E_{\mathrm{flt}}$, respectively). In the case of floating ice, it is assumed that no shearing occurs and so $E_{\mathrm{ref}} = E_{\mathrm{flt}}$ here. For grounded ice, first the fraction of the effective strain rate attributed to vertical shearing is calculated as:

$$f_{\mathrm{z}} = \frac{\left(\dot{\varepsilon}_{\mathrm{xz}}^2 + \dot{\varepsilon}_{\mathrm{yz}}^2\right)}{\dot{\varepsilon}^2}, \tag{4}$$

where $\dot{\varepsilon}_{\mathrm{xz}}$ and $\dot{\varepsilon}_{\mathrm{yz}}$ are the shear-strain components of the strain rate tensor, and $\dot{\varepsilon}$ is the effective strain rate. Note that $f_{\mathrm{z}}$ is a 3D diagnosed field. The reference enhancement factor for grounded ice is then calculated as the weighted mean between the shear and stream parameter values:

$$E_{\mathrm{ref}} = f_{\mathrm{z}} E_{\mathrm{shr}} + (1 - f_{\mathrm{z}}) E_{\mathrm{strm}}. \tag{5}$$

We set $E_{\mathrm{strm}} = 1.0$ and $E_{\mathrm{flt}} = 0.7$ following Ma et al. (2010), while $E_{\mathrm{shr}}$ is kept as a free parameter for the model evaluation. Meanwhile, $E_t$ is set to 1 for floating ice and ice streams, as any transient evolution of the ice fabric is expected to break down in these regions as explained above. Ice streams are defined in this context as grounded ice with a velocity magnitude greater than $100\,\mathrm{m\,yr}^{-1}$. For grounded ice flowing at speeds below this threshold, $E_t$ is treated as a conservative tracer. The surface boundary condition varies in time and is prescribed as:

$$E_t (z = z_s) = \alpha_e E_{\mathrm{glac}} + (1 - \alpha_e) E_{\mathrm{int}} \tag{6}$$

where $E_{\mathrm{int}}$ and $E_{\mathrm{glac}}$ represent the prescribed interglacial and glacial enhancement factors, and $\alpha_e$ is the glacial index shown in Fig. 1. $E_t$ is defined relative to $E_{\mathrm{ref}}$, in that we set $E_{\mathrm{int}} = 1$ and leave $E_{\mathrm{glac}}$ as a free parameter. $E_t$ thus acts as an amplifier to $E_{\mathrm{ref}}$. The 3D conservative enhancement factor tracer $E_t$ is determined using a second-order, upwind Eulerian tracer scheme.

The resulting enhancement factor field reflects our expectations based on observations. Floating and streaming ice regimes are relatively stiff, while ice in shearing regimes is softer when it is glacial-age ice and undergoing strong shear.

## 2.2 Paleoclimate boundary conditions and surface mass balance

The climate is calculated following a classical snapshot method, in which the present-day climate (PD) and that of the last glacial maximum (LGM) are known (e.g., Greve et al., 1999; Marshall et al., 2000). The climatic forcing for other times is an





interpolation of the two snapshots with weighting following a glacial index. The near-surface air temperature field $T$ is thus calculated as

$$T = T_{\mathrm{pd}} + \alpha_c \left( T_{\mathrm{lgm}} - T_{\mathrm{pre}} \right), \tag{7}$$


where $T_{\mathrm{pd}}$ is the present-day climatology obtained from a regional climate model simulation, $T_{\mathrm{lgm}}$ and $T_{\mathrm{pre}}$ are results from climate model simulations of the LGM and preindustrial period, respectively, and $\alpha_c$ is the glacial climate index shown in figure 1. Precipitation is calculated analogously,

$$P = P_{\mathrm{pd}} \left( \alpha_c \left[ \frac{P_{\mathrm{lgm}}}{P_{\mathrm{pre}}} - 1 \right] + 1 \right) + \alpha_p \Delta P_{\mathrm{hol}}, \tag{8}$$

although the anomaly scaling is applied as a ratio rather than a sum to avoid negative values. An additional index, $\alpha_p$ (Fig. 1c), is used to impose a spatially-constant precipitation anomaly only during the Holocene using the parameter $\Delta P_{\mathrm{hol}}$. This term adds a degree of freedom to the precipitation, which represents atmospheric changes not captured by the available snapshots. The temperature precipitation fields are additionally scaled to be consistent with the dynamic ice-sheet topography via a fixed lapse rate of -6.5 K/km.

The glacial climate index $\alpha_c$ used here is a hybrid of several reconstructions of the Greenland temperature anomaly that span the time period of interest (Vinther et al., 2009; Barker et al., 2011; Dahl-Jensen et al., 2013; Kindler et al., 2014). The individual reconstructions are scaled to match temperature anomalies reconstructed for the NGRIP ice core (Kindler et al., 2014), and then a low-pass filter is applied to remove noise at time scales under 10 kyr. Finally, the time series is normalized to give $\alpha_c(\mathrm{PD}) = 0$ and $\alpha_c(\mathrm{LGM}) = 1$. The glacial index is applied to monthly climate data. Figures 2 and 3

show the annual-mean, near-surface air temperature and precipitation for PD and LGM. The PD snapshot was obtained as the 1981-2010 climatic average of a simulation of the regional climate model MAR (Fettweis et al., 2008, 2017) forced at the boundaries by the ECMWF Interim Reanalysis (Dee et al., 2011). In the case of temperature, the LGM snapshot is the ensemble mean of simulations contributed to the Paleoclimate Modelling Intercomparison Project Phase III (PMIP3) from several participating models (CCSM4, CNRM-CM5, FGOALS-G2, GISS-E2-R, IPSL-CM5A-LR, MIROC-ESM, MPI-ESM-

P, MRI-CGCM3) (Abe-Ouchi et al., 2015). Given the large uncertainty associated with precipitation, we sample the mean $\bar{P}_{\mathrm{lgm}}$ and standard deviation $\sigma_P$ of the PMIP3 simulations to obtain the LGM precipitation snapshot for a given simulation as:

$$P_{\mathrm{lgm}} = \bar{P}_{\mathrm{lgm}} + f_{\mathrm{LGM}}\, \sigma_P, \tag{9}$$

where $f_{\mathrm{LGM}}$ is a free parameter used to scale the uncertainty around the mean. The mean and and standard deviation of the LGM precipitation fields are shown in figure 3.

Once the monthly temperature and precipitation fields are known for a given time, the surface mass balance is calculated using the positive degree day (PDD) method (Reeh, 1991). Temperatures above the freezing point are converted into melting





of snow and ice via the parameters $\beta_s$ and $\beta_i$, respectively. Here $\beta_i = 7\ \mathrm{mm\,K^{-1}\,day^{-1}}$ and $\beta_s$ is kept as a free parameter. The ice surface temperature is calculated as $T_s = \min\left(T_{\mathrm{ann}} + 0.0266\dot{m}_s, 273.15K\right)$ where $T_{\mathrm{ann}}$ is the mean annual near-surface air temperature and $\dot{m}_s$ is the net melt rate at the surface. $\dot{m}_s = 0\ \mathrm{m\,yr^{-1}}$ when no melting occurs or all melt is refrozen in the

snowpack.

Marine-shelf melting is calculated following previous work (Tabone et al., 2018), with the basal mass balance of floating ice calculated as:

$$\dot{b}_{\mathrm{shlf}} = \dot{b}_{\mathrm{ref}} - \kappa\Delta T_{\mathrm{shlf}}. \tag{10}$$

Here $\dot{b}_{\mathrm{ref}} = -10\ \mathrm{m\,yr^{-1}}$ is a spatially constant shelf basal mass balance representing the PD rate and $\kappa$ is a heat-flux

coefficient that translates oceanic temperature anomalies into basal melting. Values of $\kappa=10\ \mathrm{m\,yr^{-1}\,K^{-1}}$ and $\kappa=1\ \mathrm{m\,yr^{-1}\,K^{-1}}$ are used for shelf-ice near the grounding line and the broader shelf, respectively. The oceanic temperature anomaly relative to PD, $\Delta T_{\mathrm{shlf}}$, is calculated as a fraction of the atmospheric temperature anomaly: $\Delta T_{\mathrm{shlf}} = 0.25\left(T - T_{\mathrm{pd}}\right)$ (Golledge et al., 2015). $\dot{b}_{\mathrm{shlf}}$ is limited to negative values (i.e., melting) to prevent unrealistic rates of ice accretion during cold climates.

At the lower boundary, the geothermal heat flux $Q_{\mathrm{geo}}$ is imposed with a spatially constant value. It too is a free parameter.

Isostatic rebound is calculated dynamically using an Elastic Lithosphere Relaxing Aesthenosphere (ELRA) isostasy model with a spatially-constant time constant set to 3 ka (Ritz et al., 1997). Sea level is spatially constant and varies in time following the global glacial cycle simulation of Ganopolski and Calov (2011) (Fig. 1).

## 2.3    Spin up

The spin up procedure consists of two steps: a temperature spin-up and optimization of the basal friction coefficient field. First,

the ice sheet is simulated under constant LGM climatic conditions for 20 kyr using the SIA ice dynamics solver alone, which is mainly intended to spin-up the ice temperature field. Next, the ice sheet is run for another 10 kyr with the full hybrid ice dynamics active. The $c_b$ field used here initially has been tuned to give good results in ice thickness for a steady-state present day simulation. By the end of this 30 kyr spin up, the ice sheet extends to near the continental shelf break, in good agreement with the reconstructed LGM ice extent (Lecavalier et al., 2014).

Second, transient simulations from the LGM to PD are performed iteratively, with the basal friction field $c_b$ modified at each iteration to improve the simulated present-day ice thickness following Pollard and DeConto (2012). When the simulation reaches PD, the ice thickness error is calculated ($H_{\mathrm{err}} = H_{\mathrm{sim}} - H_{\mathrm{obs}}$). For each grid point, we then use the upstream $H_{\mathrm{err}}$ to calculate a tuning factor:

$$\varepsilon = \frac{H_{\mathrm{err}}}{H_0}, \tag{11}$$

where $H_0 = 1000\ \mathrm{m}$ is a scaling parameter controlling the magnitude of changes to $c_b$ for a given iteration. For a given location, $H_{\mathrm{err}}$ is defined as the velocity-weighted average of the upstream values in both lateral dimensions. Also, we limit

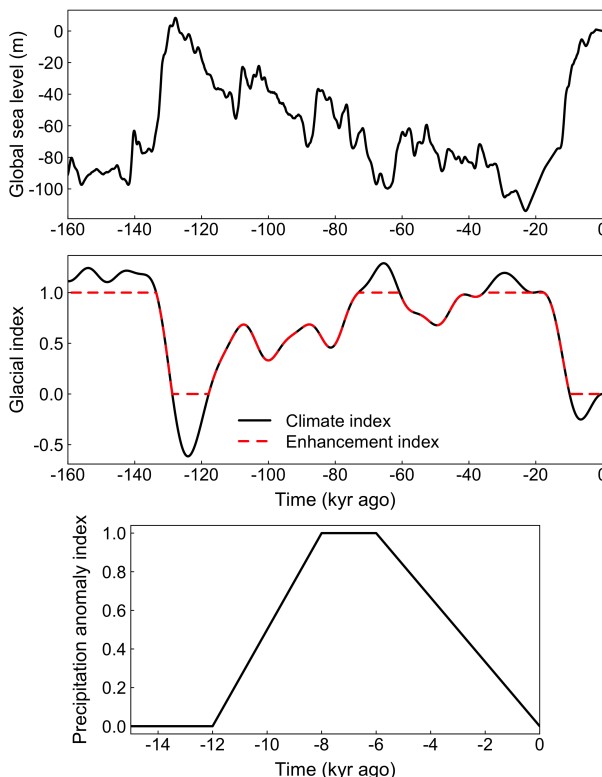

**Figure 1.** Forcing applied to the simulations. Top panel: global sea-level anomaly relative to present day. Middle panel: Glacial indices derived from reconstructed paleo temperature anomalies over Greenland. The glacial climate index is used to interpolate between the LGM climate snapshot and the present-day climate. The glacial enhancement index is used to interpolate between the imposed enhancement factor for ice during glacial periods and interglacial periods. Bottom panel: Holocene precipitation anomaly index, scales the imposed $\Delta P_{\mathrm{hol}}$ parameter in time to ensure the maximum Holocene precipitation anomaly occurs during the mid-Holocene optimum.

$\varepsilon \in (-1.5, 1.5)$ in order to avoid scaling $c_b$ too rapidly for large values of $H_{\mathrm{err}}$. The tuning factor is then used to modify the local basal friction coefficient as:

$$c_b{}^{n+1} = 10^{-\varepsilon} c_b{}^n, \tag{12}$$

where the indices $n$ and $n+1$ indicate the current and next iteration. Thus, a positive bias in upstream ice thickness $H_{\mathrm{err}}$ results in a reduction of the basal friction coefficient $\beta$, which in turn lead to a lower basal velocity for the same basal shear stress (eq. 2). Once the $c_b$ for the next iteration has been calculated, the model state is reset to the LGM spin up and the transient simulation is repeated. We performed 10 iterations, after which the error in simulated PD tends not to reduce further and the $c_b$ solution is rather stable. The above procedure results in a reasonable internal temperature distribution in the ice sheet that




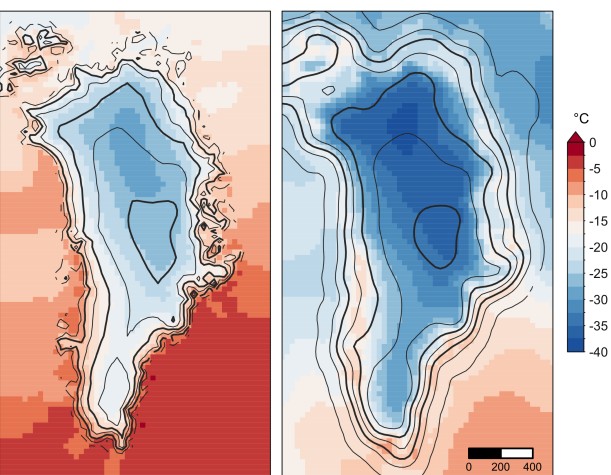

**Figure 2.** Climate snapshots. Near-surface mean annual air temperature field imposed for present-day (left) and LGM (right) conditions. Present-day data is from a regional climate model simulation (MAR v3.9), averaged over the period 1981-2010. LGM temperatures are the PMIP3 average.

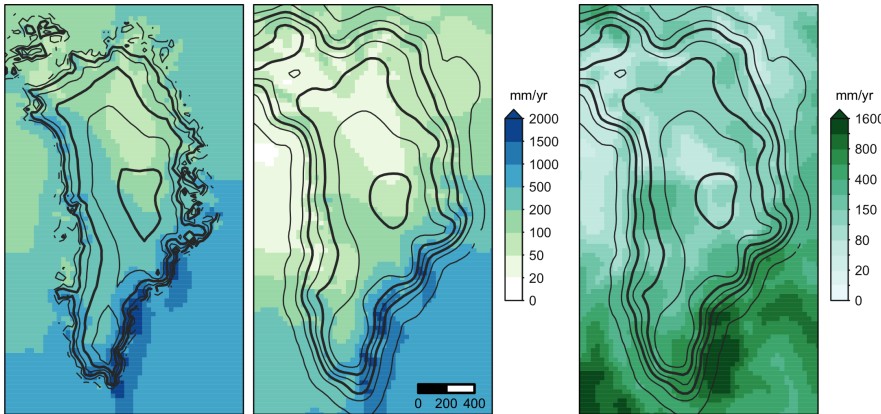

**Figure 3.** Climate snapshots. Mean annual present-day (left) and LGM (middle) precipitation snapshots, along with the uncertainty in the LGM precipitation (right). Present-day climate is the 1981-2010 average from the regional climate model (MARv3.9). LGM precipitation and uncertainty are the mean and standard deviation of model results contributed to the PMIP3 database.





is consistent with the optimized basal friction configuration. This spin-up procedure is applied to each model version in the ensemble method described below.

## 2.4    The isochronal layer tracing scheme

The tracing of isochronals is based on Born (2017). Key to the successful simulation of the isochronal surfaces is a vertical grid that is defined in time, not space, to avoid vertical advection across grid interfaces and therefore eliminate numerical diffusion.
The spacing of this isochronal grid is arbitrary and can in principle be variable, but here we use a constant resolution of 200 years. This means that the domain is virtually empty at the beginning of a simulation, aside from a few initialization layers. They are inconsequential for the analysis below as they can be regarded as older than 160 kyr and become very thin at the present day. Over the course of the simulation, one additional layer of ice is added every 200 years until the grid is full at the end of the simulation period. The horizontal grid of the layer tracing scheme is the same as in Yelmo.

A major difference to the original implementation is that the calculation of the ice physics and all boundary conditions are now being handled by Yelmo. The host model provides the two horizontal velocity components, the total ice thickness, and the mass fluxes at the ice surface and the bed to the tracer scheme at every tracer time step, $\Delta t = 5$ yr. The ice velocities are vertically interpolated to the isochronal grid using the vertically integrated layer thicknesses of the latter as reference. This can be done safely because velocities vary smoothly in the vertical dimension. As in the original formulation, layer thickness is a
passive tracer variable that is advected using an implicit upstream scheme. While the ice thickness is not strictly needed, it is used here to ensure that the sum of isochronal layer thicknesses remains consistent with Yelmo by periodically normalizing their values. This correction is minor as its only purpose is to counteract numerical drift.

   Mass fluxes at the upper and lower boundaries are applied to the tracer scheme at the temporal resolution of the isochronal grid. Between these times, the surface and bottom mass balances of Yelmo are integrated so that the amount of ice in each new
layer depends on the cumulative surface mass balance of the previous 200 years. Where the mass balance is negative, also at the bed, the appropriate amount is removed from the next available layers. Thus, older layers naturally outcrop at the surface near the margins.

## 2.5    Ensemble design

An ensemble of 300 simulations was performed to investigate the impact of model and experimental choices on the simulation
of isochronal layers. Each simulation consists of the spin up and $c_b$ optimization procedure described above, followed by simulations of the ice sheet from 160 kyr ago to PD with fully transient boundary conditions. Four model parameters, the PDD factor for snow $\beta_s$, the geothermal heat flux $Q_{geo}$, and the flow enhancement factors for shearing $E_{shr}$ and glacial ice $E_{glac}$, and two boundary-forcing parameters, the Holocene precipitation anomaly $\Delta P_{hol}$ and the glacial precipitation anomaly $f_{LGM}$, were perturbed using a Latin-Hypercube sampling approach.



## 3 Results


The skill of the ensemble simulations is quantified by the root mean square error (RMSE) of the ice thickness and the depth of four key isochrones below the surface. The age of these selected isochrones was chosen by the authors of the original dataset (MacGregor et al.) and their dating is based on a combination of ice core data where they intersect radar reflections as well as the quasi-Nye method. The isochrone data and their uncertainty was then interpolated to the horizontal Yelmo grid. Data

uncertainty is not taken into consideration when calculating the RMSE but this is usually small compared to the data-model mismatch.

In addition to evaluating the ice surface and isochrone depths individually, we also calculate the RMSE across all five evaluation surfaces together as a metric of overall simulation quality. It should be noted that because of the incomplete spatial coverage of the englacial data, ice thickness has a relative higher weight in the combined RMSE. About 39% of all points in

the combined RMSE are part of the ice thickness field, nearly double of what may be assumed based on it being one of five fields. Only the region with an ice thickness of more than 1000 m is considered to avoid contamination from areas near the ice margin where Yelmo cannot adequately resolve the topography and ice dynamics due its relatively coarse resolution (black contour, Fig. 4).

The presentation of the results is structured to provide a qualitative overview by discussing two examples from the ensemble

in section, followed by a detailed discussion of the full ensemble in section 3.2. The two example simulations are the one with the lowest RMSE for the ice thickness, $BEST_{ice}$, and the one with the best skill for the combined RMSE of all five surfaces taken together, $BEST_{all}$, to illustrate the benefits of including englacial data.

### 3.1 Simulation of isochrone depth

The simulated ice thickness and isochrone depths are in qualitatively good agreement with observations in both $BEST_{ice}$ and

$BEST_{all}$, but closer inspection reveals important differences. $BEST_{ice}$ agrees with the observations of ice thickness within a few tens of meters, but also shows large disagreements in the depth of the isochronal surfaces (Fig. 4). The contrary is true for $BEST_{all}$ (Fig. 5). The anomaly pattern of ice thickness of both simulations shows the expected deviations near the ice margin and a mostly homogeneous pattern in the ice sheet interior that does not point to any systematic bias. The anomalies in isochrone depths do show a more detailed pattern of positive and negative anomalies in $BEST_{all}$. The positive anomaly in the

northwest is visible in all four isochrones and may indicate too much accumulation in the simulation or a too rapid ice flow in this region that includes the ice divide, or both. Similarly, the negative anomaly in the northeast may be interpreted as too little accumulation or a dynamic bias in the model, potentially associated with the Northeast Greenland Ice Stream.

A section through the summit confirms that $BEST_{ice}$ more closely captures the correct ice thickness but that the englacial stratigraphy is in worse agreement with observations than in $BEST_{all}$ (Fig. 6). However, looking at the absolute elevation of the

isochrones instead of its depth below the surface, it is clear that the relatively good agreement in isochrone depth in $BEST_{all}$ is largely due to an underestimated ice thickness. This is an interesting result because it suggests that future improvements should focus on the early part of the simulation. We also observe that despite these shortcomings, $BEST_{all}$ is among the best


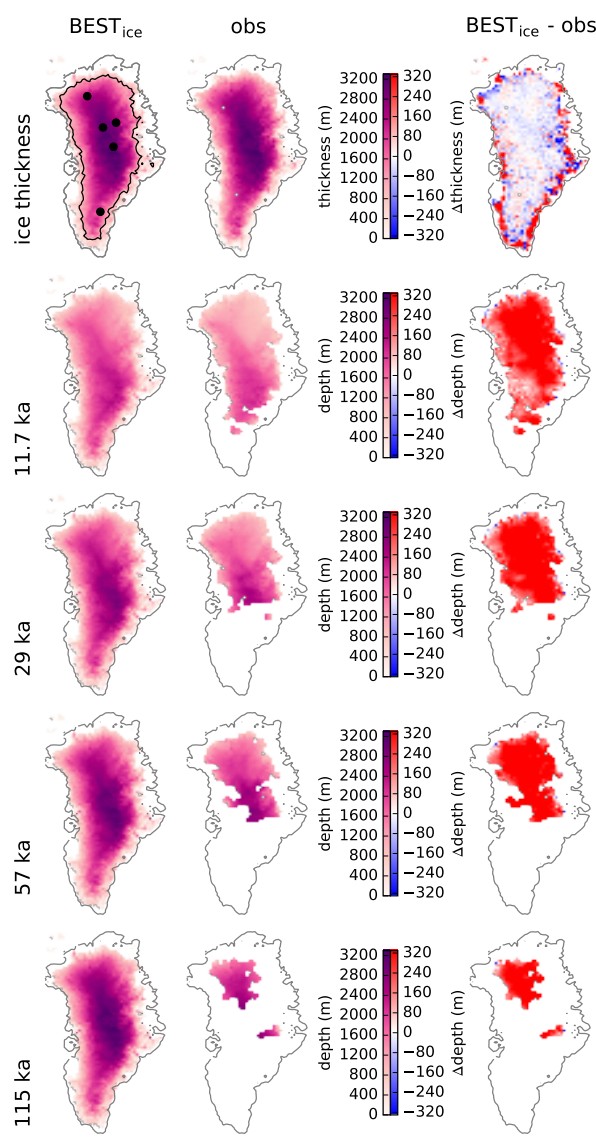

**Figure 4.** Ice thickness and isochrone depth for the simulation BEST$_{ice}$, observations, and the respective differences. Top left panel shows the 1km ice thickness contour and selected ice core locations, from north to south: NEEM, EGRIP, NGRIP, Summit, Dye-3.

simulations in simulating the total ice thickness while BEST$_{ice}$ show a low skill in the combined RMSE (Fig. 7). Similarly, the four simulations with best results for the individual isochrones (indeed four different ones) have reasonably low values for
the RMSE of total ice thickness. In contrast to this, the simulation with the best ice thickness performs poorly in all isochrone depths.

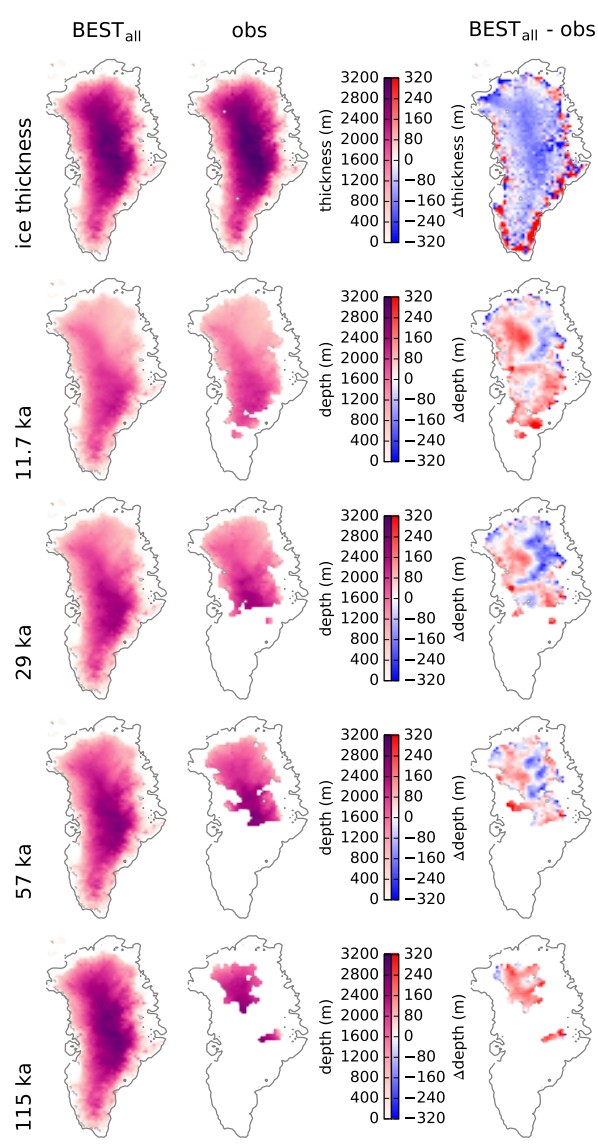

**Figure 5.** As figure 4, but for simulation BEST$_{\mathrm{all}}$.

## 3.2 Sensitivity to tuning parameters

Not all of the six tuning parameters contribute equally to the ensemble spread (Fig. 7). The Holocene precipitation anomaly $\Delta P_{\mathrm{hol}}$ appears to have the largest impact in terms of RMSE for all control surfaces, followed by the scaling factor of the glacial precipitation anomaly $f_{\mathrm{LGM}}$. The optimal ice thickness is simulated with relatively high values for $\Delta P_{\mathrm{hol}}$, but high Holocene precipitation deteriorates the simulated depth of the isochrones. This appears to be the primary reason behind the

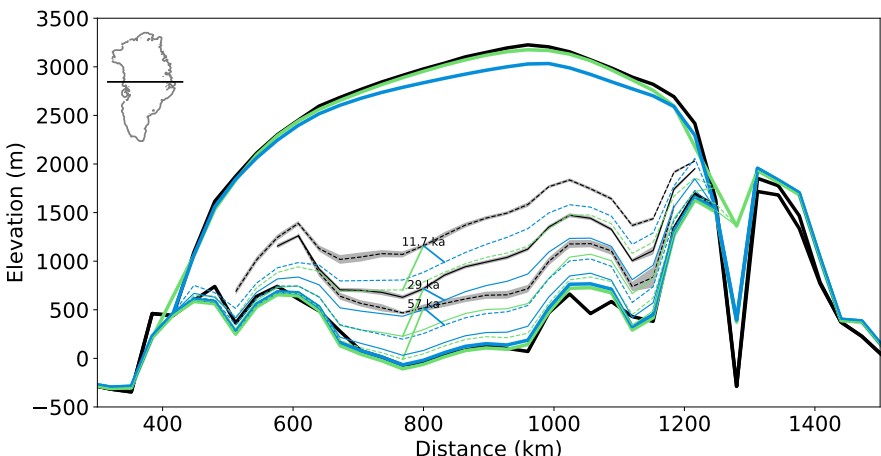

**Figure 6.** Section through the summit, showing ice surface and bedrock topographies (bold lines), and isochrones (thin lines), each for observations (black), BEST$_{ice}$ (green), and BEST$_{all}$ (blue). Isochrones alternate between solid and dashed line style to improve visibility. Diagonal lines connect corresponding isochrones for the simulations and observations. Reconstruction data for the 115 ka isochrone is not available at this location. Shading is the uncertainty in the observations.

large differences between BEST$_{ice}$ and BEST$_{all}$. Lower values of $\Delta P_{hol}$ also worsen the RMSE of isochrone depth, so that the optimal value overall is close to 0 m yr$^{-1}$. All isochrones respond similarly to $\Delta P_{hol}$, because the anomalous precipitation increases their depth similarly. The clear minimum of RMSE in the middle of the parameter range suggests that it was chosen
appropriately although the use of a constant offset for all locations has its shortcomings as will be discussed below.

The scaling of the glacial precipitation uncertainty $f_{LGM}$ has the strongest impact on the 57 ka and 115 ka isochrone depths in that both benefit from lower values. The same tendency, albeit weaker, is visible for the 29 ka isochrone depth. The reduced importance is probably due to the relatively short period during which the anomalous precipitation can impact this isochrone. Glacial precipitation has only negligible influence on the 11.7 ka isochrone depth and the total ice thickness. This is to be
expected because glacial ice makes up a rather small portion of the total ice thickness.

The influence of the PDD melt factor for snow $\beta_s$ is mostly limited to the 115 ka isochrone, because the preceding period, the last interglacial, is the only time when significant melting occurs in the region above 1000 m. The geothermal heat flux $Q_{geo}$ and the two enhancement factors $E_{shr}$ and $E_{glac}$ have only minor effects. One possible explanation is that these three parameters are most important for ice dynamics that primarily take place near the bed and the ice margins. The deliberate exclusion of the
margins but more importantly the very limited coverage of observational data for the deeper isochrones may bias the results to regions where dynamic thinning is least pronounced. In addition, the basal friction optimization may counteract the effects of changing the dynamic parameters, because it is carried out for every combination individually. Lastly, the relatively strong impact of the precipitation parameters $\Delta P_{hol}$ and $f_{LGM}$ can potentially dominate and therefore mask the effects of the other parameters.





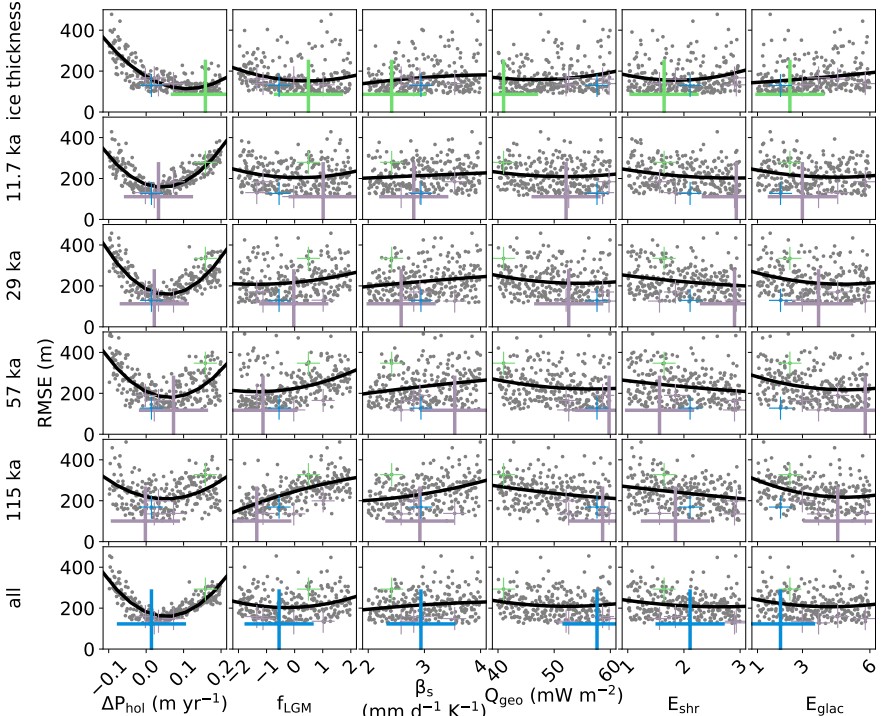

**Figure 7.** RMSE of ice thickness and isochrone depth for the six tuning parameters. The bottom row shows the RMSE of all isochrones and the total ice thickness together. Individual simulations are marked with gray dots, large crosses mark the location of the best simulation for each metric as indicated on the left. Smaller crosses of the same color mark the location of the same simulations in the other metrics. A black line illustrates a second order polynomial fit.

The relatively low sensitivity of some of the parameters and the large scatter around the trend lines mean that several simulations across the parameter range produce RMSEs that are comparable to $BEST_{ice}$ and $BEST_{all}$. As a corollary to this finding, although these two sets of parameters produce the optimal results in our ensemble, the parameters themselves are not incontrovertibly optimal (Fig. 8). The best ten percent of simulations agree well on $\Delta P_{hol}$ and $f_{LGM}$, but otherwise span a large part of the tested parameters range.

Additional information can be obtained from the vertical distance between isochrones, not just their depth below the surface (Fig. 9). $\Delta P_{hol}$ does not have a direct effect on these differences and so they allow inferences on the dynamic response. A higher Holocene precipitation generally has a slight positive effect, in particular on the deepest isochrones, suggesting that they benefit from stronger dynamic thinning. Interestingly, all isochrones agree that a high value for $\Delta P_{hol}$ would be detrimental to their simulated depth below the surface (Fig. 7). We therefore conclude that the imperfect thickness of the layer between the 115 ka

and 57 ka isochrones is due to an unrelated model bias such as an unrealistic SMB during that time interval or shortcomings in the simulated dynamics. $\Delta P_{hol}$ may partially counteract these by enhancing dynamic thinning, but at the risk of fortuitously

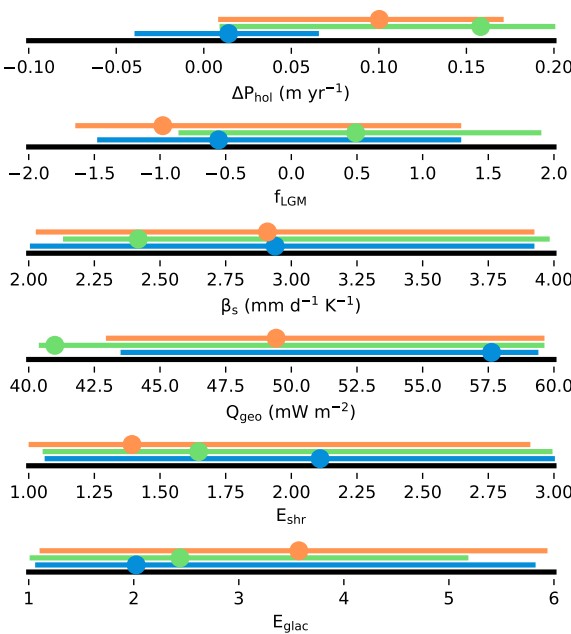

**Figure 8.** Parameter range for the top 10% simulations for the ice thickness metric (green), the combined RMSE (blue) and the combined RMSE based on the Eulerian age tracer (orange). Simulations BEST$_\mathrm{ice}$, BEST$_\mathrm{all}$, and the corresponding optimal simulation for the Eulerian age tracer are highlighted with dots.

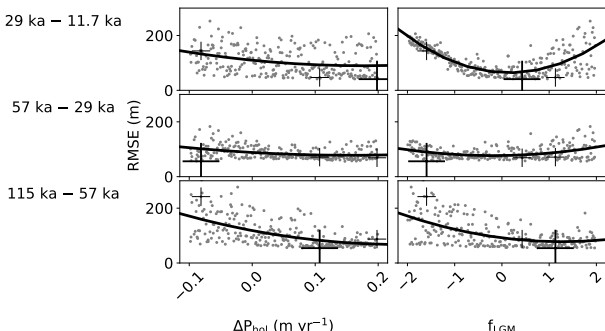

**Figure 9.** As figure 7, but for differences between isochrones. Note the different scale on the vertical axis.

combining two errors into one seemingly correct outcome. A similar conclusion can be drawn from the decrease in RMSE of the 115-57 ka difference with higher $f_\mathrm{LGM}$. However, the thickness of the 29-11.7 ka layer, which is directly impacted by the change in glacial precipitation, clearly favors a small value for this parameter.




The RMSE does not provide information on the sign or the spatial pattern of the disagreements. To address this, we define composites for the five evaluation surfaces that represent high and low values of the parameter range. This is only shown for the two most sensitive parameters $\Delta P_\mathrm{hol}$ and $f_\mathrm{LGM}$ (Fig. 10 and 11). Both figures are separated into a comparison of the composites with the ensemble average to better understand the sensitivity of the evaluation surfaces to the parameters (left) and with observations (right).

Although high values of $\Delta P_\mathrm{hol}$ make the GrIS thicker, the simulated ice thickness is always below the observed. The comparison of the high $\Delta P_\mathrm{hol}$ composite with the ensemble average shows that the increase in precipitation has its largest impact in the northern part of the ice sheet. Because this region is the dryest, the constant offset of $\Delta P_\mathrm{hol}$ makes the largest relative difference here. As a consequence, isochrone depths show the largest anomalies in the northern and northeastern part as well when compared with the ensemble average.

Comparison of the isochrone depth with observations shows that all isochrones are too deep for the high composite and too shallow for the low $\Delta P_\mathrm{hol}$ composite, indicating that the parameter range was chosen adequately. The anomalies of the low $\Delta P_\mathrm{hol}$ composite with respect to observations have a distinct shape over the north and along the northeastern half of the GrIS. This was also seen in the comparison of the low composite with the ensemble average, but no equivalent is found when comparing the high composite with the observations. This results indicates a nonlinear response to anomalous Holocene

precipitation, probably due to the flow of ice. Unfortunately, isochrone data from the south is sparse and does not contribute to constraining $\Delta P_\mathrm{hol}$. The difference of the ensemble average and observations shows a pattern of both positive and negative anomalies. This may be interpreted as shortcomings in the precipitation data that was used to force the simulations. The comparison with isochrone data could provide a way of addressing this issue in the future.

Regarding the vertical distance between isochrones, a higher Holocene accumulation minimally increases the thickness of

the $29 - 11.7$ ka layer (Fig. 10), because $\Delta P_\mathrm{hol}$ sets in before 11.7 ka (Fig. 1). The older isochrones are all pushed closer together by enhanced dynamic thinning. This effect is especially obvious with anomalously low $\Delta P_\mathrm{hol}$, where the increased isochrone differences in the center are contrasted by a closer vertical distance at the margins because less ice is advected there. In general, the response of the isochrone differences is much less linear than that of their depth below the surface, as seen in the distinct differences between the high-ave and low-ave differences. The reason for this is that the distance between isochrones

that are not directly affected by changes in Holocene accumulation are wholly due to changes in ice dynamics that are known to be highly nonlinear. In contrast, the depth below the surface is a direct and linear consequence of higher precipitation. The comparison of the vertical isochrone distance with observations reveals a heterogenic pattern that does not allow for a robust interpretation at this point.

The composite fields for $f_\mathrm{LGM}$ show that the ice sheet becomes thicker with increasing precipitation during the glacial

period, but as for $\Delta P_\mathrm{hol}$ the increase is not enough to achieve a realistic ice thickness (Fig. 11). The thickening is also limited to the center of the GrIS while the thickness decreases around the margins. This is because the higher accumulation also increases ice flow and therefore the removal of mass. This is especially evident from the depth of the 11.7 ka isochrone that becomes noticeably shallower with high $f_\mathrm{LGM}$. Holocene ice that accumulates on an ice sheet that experienced higher glacial accumulation will experience a thicker ice sheet with steeper slopes, which more efficiently removes new ice. The





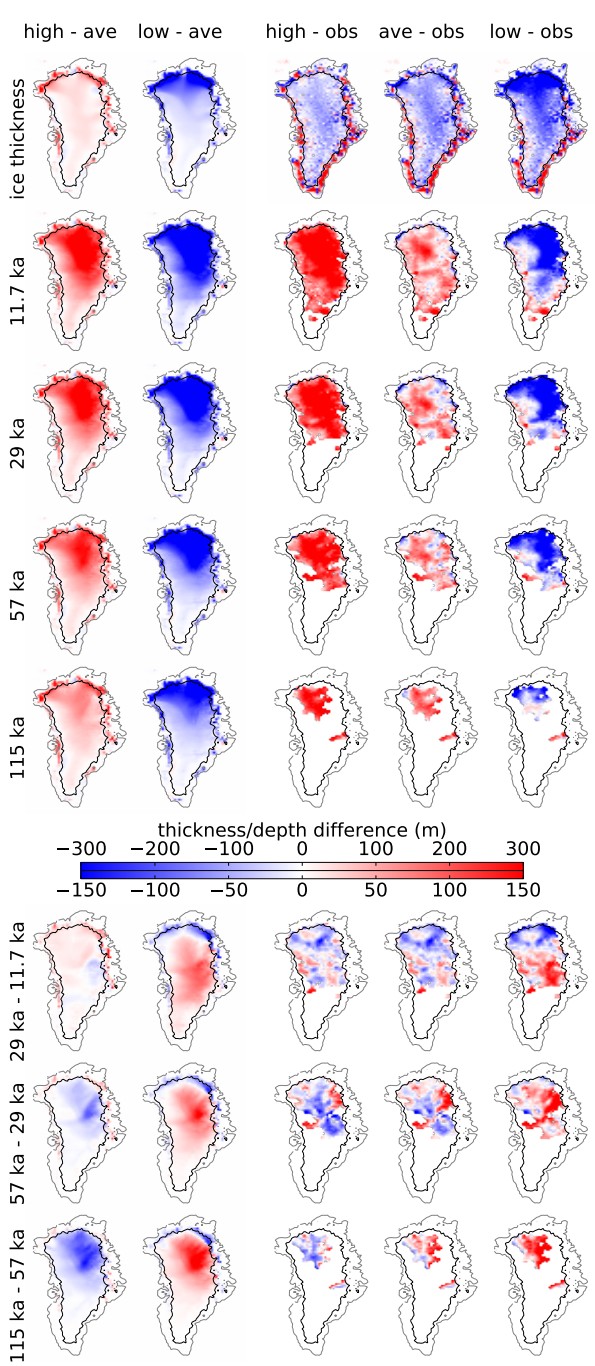

**Figure 10.** Composite ice thickness and isochrone depth for simulations with high (-0.1 to -0.05 m yr$^{-1}$) and low (0.15 to 0.2 m yr$^{-1}$) values of $\Delta P_{\text{hol}}$, compared to the ensemble average and observations. A thin black line marks the 1000 m surface elevation.



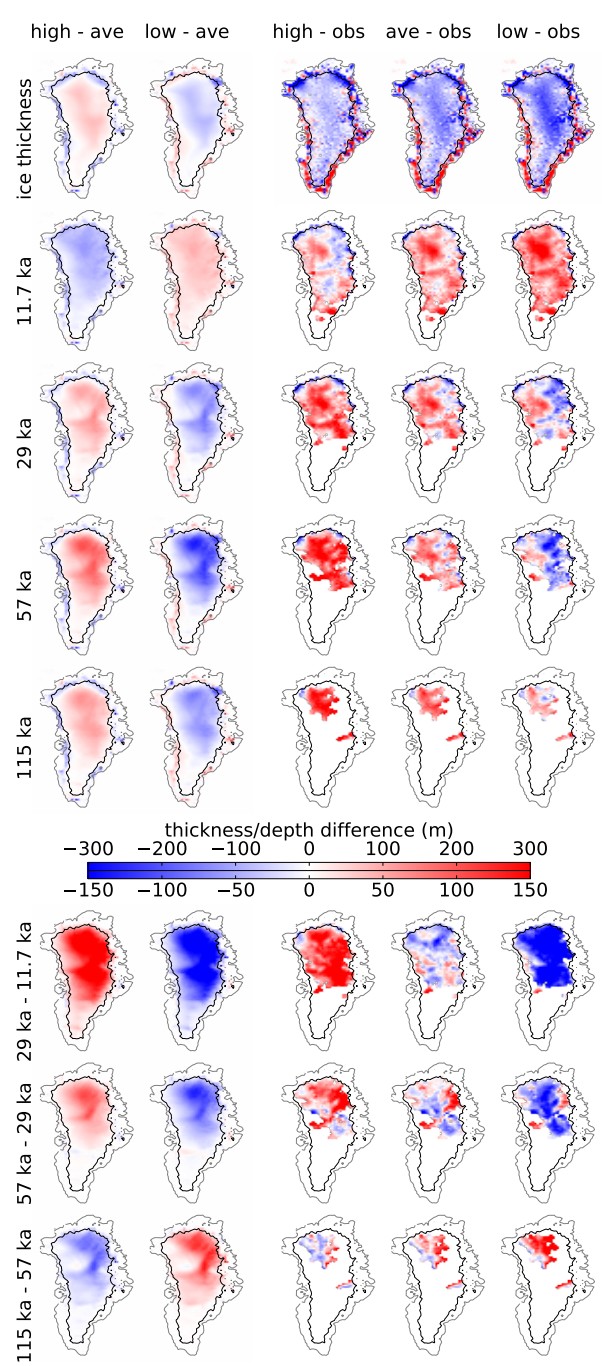

**Figure 11.** As figure 10, but for composites of variable $f_{\mathrm{LGM}}$. The ranges of the high and low composites are 1.5 to 2 and -2 to -1.5.



older isochrones all experience the direct effect of higher glacial precipitation and their depth below the surface therefore is invariably higher for larger values of $f_{\mathrm{LGM}}$. The pattern of the precipitation anomaly field can be recognized in the isochrone depths (Fig. 3).

The difference of the low $f_{\mathrm{LGM}}$ composite for 29 ka and 57 ka isochrone depths and the corresponding observations shows the fingerprint of the precipitation anomaly pattern (Fig. 3). There is a negative anomaly in the northeast while the reduc-

tion of glacial precipitation is not enough to eliminate the positive isochrone depth anomaly in northwestern Greenland. The pre-Holocene isochrones of the high $f_{\mathrm{LGM}}$ composite are generally too deep (ave–obs), suggesting a negative $f_{\mathrm{LGM}}$ as an improvement, but the imposed glacial precipitation anomaly pattern is not optimal to address spatial heterogeneities. This suggests that this precipitation anomaly pattern is a poor match for the unperturbed model's shortcomings, which is not wholly unexpected because it only represents climate model disagreement.

The vertical distance between isochrones increases with $f_{\mathrm{LGM}}$ for glacial periods, while the oldest layer, 115 ka – 57 ka, records the dynamic thinning of the additional load overhead. The positive and negative composites are mostly symmetric when compared with the ensemble average. As for the comparison with observations, clear positive and negative anomalies are found for the high and low composites, so that the parameter range is likely chosen adequately for this particular metric.

### 3.3 Simulated depth profiles, ice volume, and comparison with Eulerian age tracer

Where most of the above analysis concerned a spatially comprehensive view of only four isochronal surfaces, the high resolution of our model in the temporal domain allows for a different and complementary perspective, the full age profile at certain locations (Fig. 12).

All locations show progressive thinning toward the bed, which is more pronounced for regions of high accumulation as expected. Ice is too young in most of the ensemble simulations, probably due to excessive accumulation, in particular at the

more southerly locations Dye-3 and Summit. A less severe mismatch is found for NGRIP where the radiostratigraphy age profile almost coincides with the 25%/75% percentiles envelope of the ensemble, while a much better agreement is found at NEEM and EGRIP. The most plausible explanation for the difference in simulation quality between north and south are deficiencies in the surface mass balance due to the uncertain climate of the past or the relatively simple parameterization of mass balance used here. However, it is conceivable that the lack of observational isochrone data in the south accentuates these

issues or adds to them. For example, the better coverage of radiostratigraphy data in the north biases the PDD melt factor $\beta_{\mathrm{s}}$ toward this region and its climatic conditions, which likely is a poor choice for the climatically very different south. A similar effect is possible for the parameters that control ice dynamics, that may be different for the fast-flowing ice of the south.

BEST$_{\mathrm{ice}}$ has much younger ice everywhere in the ice column due to its unrealistically high accumulation during the Holocene and glacial period. BEST$_{\mathrm{all}}$ achieves a good simulation of the age profiles at NEEM and EGRIP, including periods of rapid age

increase around 29ka and between 60ka and 70ka that agree well with the radiostratigraphy data at the EGRIP site and less so at NEEM.

Comparison with the Eulerian age tracer from the same simulation, BEST$_{\mathrm{all}}$, shows a clear disagreement with the age profile simulated by the isochronal tracing scheme, although they should ideally be identical because they simulate the same variable





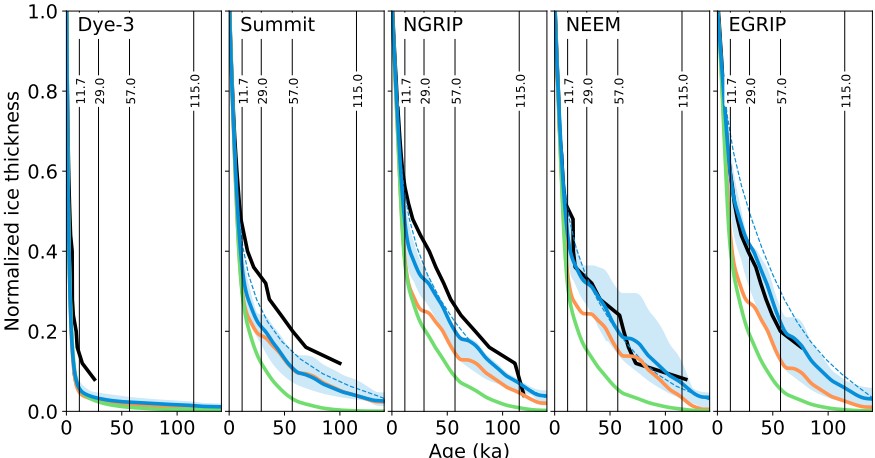

**Figure 12.** Age profiles of selected ice core locations (see Fig. 4 for map). The curves show radiostratigraphy data (black), BEST$_{ice}$ (green), BEST$_{all}$ (blue), the optimal simulation for the Eulerian age tracer (orange), and the Eulerian age tracer of BEST$_{all}$ (dashed blue). The shading are the 10% best simulations in the combined RMSE.

using two different methods. We observed that the Eulerian tracer data generally shows a weaker curvature, almost invariably

older ages, and does not capture the variations in accumulation on shorter time scales, all of which indicate that it is subject to numerical diffusion in the vertical dimension. The isochronal layer tracing scheme circumvents this problem by eliminating flow across layer boundaries and by using a numerically much less complex implementation that only requires advection in the two horizontal dimensions.

The presumed spurious bias toward older ice with the Eulerian scheme may have a noticeable effect on a calibration based

on its results. To counteract the upward diffusion of older ages, higher values for Holocene precipitation, $\Delta P_{hol}$, may seem advantageous (Fig. 8). Following the same explanation, less glacial precipitation and so lower values of $f_{LGM}$ are necessary to obtain a good match with observations. Consequently, the simulation with the lowest RMSE based on the Eulerian age tracer shows a clear bias toward older ages in the analysis of the more accurate isochronal age tracer (Fig. 12, orange). The other ensemble parameters are less well constrained both for the isochronal and the Eulerian age schemes and differences are

negligible.

The tendency toward higher Holocene and lower glacial precipitation in the Eulerian-calibrated simulation is also evident in the total simulated ice volume (Fig. 13). Simulations BEST$_{ice}$ and BEST$_{all}$ show a similar ice volume for most of the simulation period, with notable exceptions in the relatively warm Holocene and Eemian. This suggests that the type of calibration has significant impact on model sensitivity. The simulated ice volume of all simulations is low when compared to earlier re-

constructions (e.g., Lecavalier et al., 2014). This is the result of relatively low basal friction in the marine sectors surrounding Greenland and the use of a hybrid ice-sheet model, which leads to thinner grounded ice on the continental shelf. Previous ice volume estimates were based on SIA models that generally simulate thicker ice when constrained with the same ice extent.





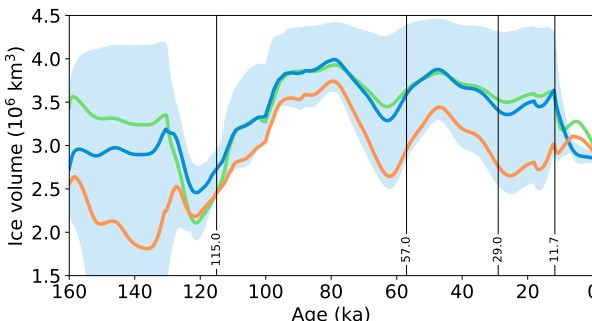

**Figure 13.** Volume of the Greenland ice sheet as simulated by BEST$_{ice}$ (green), BEST$_{all}$ (blue), the 10% best simulations in the combined RMSE (shading), and the simulation with the lowest combined RMSE based on the Eulerian age tracer (orange). Vertical lines highlight the age of the calibration isochrones.

More recent simulations using hybrid models have shown lower volumes on the order of those shown here to be equally possible (Tabone et al., 2018; Buizert et al., 2018). Regardless of this open question, the ice thickness on the continental shelf is of
negligible consequence for our comparison with the observed isochrones because they are far inland and not directly influenced by the remote ice margin during the glacial period.

## 4 Discussion and Conclusions

Our results show that including englacial layers provides useful constraints for the simulation of the GrIS. Simulations that only agree with the modern ice surface topography may produce a very unrealistic ice stratigraphy, suggesting that this calibration
objective is too narrow. Given necessary limitations in our model and ensemble design such as a relatively coarse resolution and a small set of free model parameters, an optimal agreement with both the total ice thickness and isochrone depth appears mutually exclusive. However, simulations that were selected because of their good agreement with the observed isochrone depth show a reasonable simulation of ice thickness, while the opposite is not true. The simulation that best matches the observed total ice thickness is among the worst of the entire ensemble in terms of stratigraphy. This again suggests that the
englacial stratigraphy is a more reliable evaluation metric than ice thickness alone.

We find that uncertainty in surface mass balance has a far greater impact on isochrone depth than uncertainty in ice flow parameters, albeit with important regional exceptions. This result may also be influenced by our model's limited ability to simulate the fast ice flow in narrow fjords and near the margins, as well as the poor coverage of isochrone reconstructions in these regions. In addition, the impact of different ice-dynamics parameters is reduced by the optimization procedure for basal
friction. However, it could be expected that uncertain accumulation leads to a broad range of simulated isochronal depths, since accumulation directly controls depositional layer thickness. Because the simulation of past climates and the calculation





of surface mass balance is known to be highly uncertain (van de Berg et al., 2011; Merz et al., 2014a, b, 2016; Plach et al., 2018, 2019), the simulation of isochrones thus exhibits power in constraining this boundary condition.

The sensitivity of isochrone depth to variations in geothermal heat flux is also low, but increases with isochrone age. The
oldest layers (57 ka, 115 ka) contrain $Q_{geo}$ to values on the upper end of the tested range. This is an interesting result, because the direct effect of warming the ice from below are lower ice viscosities near the bed and thus higher thinning rates for deeper layers. The spatial pattern of the disagreement with observed isochrone depth may also help to constrain spatial variations in $Q_{geo}$, but this analysis would have to be performed on simulations with the same basal friction.

Disregarding its effect on ice flow dynamics for now, the mismatch of simulated and observed isochrone depths and the ver-
tical distance between isochrones allows inferences on the accuracy of accumulation at different times. The spatial patterns of these mismatches suggest possible improvements in the distribution of precipitation and melting. Our ensemble of simulations used a spatially uniform Holocene precipitation anomaly and a glacial precipitation anomaly based on differences between global climate model simulations. In addition, we used a simple PDD scheme that calculates SMB from air temperature and total precipitation using empirical proportionality factors. We argue that these choices are justified a priori because their impact
could not yet be quantified, a fact that the comprehensive simulation of isochrones may change now. The low optimal value for $\Delta P_{hol}$ may be interpreted such that the optimal uniform precipitation anomaly for the Holocene is none at all.

Although our isochronal layer tracing scheme adds significant computational cost to the existing thermomechanical ice sheet model, this is mainly due to the much larger number of layers. The flow of mass within isochrones uses an Eulerian scheme that is nearly identical to that of the host model. The higher number of layers and, importantly, preventing flow across vertical
grid boundaries results in a notably more reliable simulation of isochrones than using an age tracer on the relatively coarse numerical grid of the host model Yelmo, as shown by our comparison of the two methods within the same simulation. This result is consistent with previous studies (Rybak and Huybrechts, 2003). As a consequence of the dissimilar simulation, the optimal parameters for BEST$_{all}$ and the simulation that best matches all five evaluation surfaces based on the Eulerian age tracer are very different for the accumulation parameters $\Delta P_{hol}$ and $f_{LGM}$. This leads to notable differences in the simulated ice
volume and sensitivity to climate change. However, the other four parameters are only weakly constrained in both methods and so both can be used to exclude the most extreme parameter values. We argue that faced with large uncertainties in boundary conditions, even the less precise Eulerian tracer is sufficiently accurate to provide results better than no constraint at all, at least for younger isochrones that are less affected by numerical diffusion. Compared to Lagrangian or semi-Lagrangian tracer advection schemes, our method avoids costly interpolation or low particle densities. It is possible to use an uneven spacing of
the isochronal grid to concentrate computational cost to key periods of interest.

In summary, the isochronal modeling framework offers a significant step forward in evaluating and calibrating ice sheet models. The method used here is not exclusive to the Yelmo model and can, due to its modest requirements from the host model, readily be adapted to most existing ice sheet models. It is also possible to run the isochronal layer scheme independent from the host model if the necessary fields of ice velocity, mass fluxes, and ice thickness are available at a suitable temporal resolution.
We believe that the direct comparison of ice sheet models with one of the best and most comprehensive glaciological archives holds great potential to both future model development and the interpretation of radiostratigraphy data. Future simulations

and in particular model calibration will need to prioritize the realism of the SMB simulation, because the large uncertainties in accumulation eclipse the impact of dynamics and our ability to constrain the corresponding parameters. We plan to use a much more reliable SMB scheme in the future (Born et al., 2019; Fettweis et al., 2020; Zolles and Born, 2020), but these

simulations would still depend on uncertain past climates. Here, the isochronal model offers an alternative that is independent from climate simulations. By comparing the simulated isochrones with their observed counterparts, information about past accumulation rates can in principle be derived using inverse modeling (e.g., Waddington et al., 2007; Nielsen et al., 2015). Lastly, future work would greatly benefit from a better coverage of the dated radiostratigraphy record in the dynamically more active southern Greenland.

*Data availability.* The supplementary material contains a self-describing data set, an example python script to read it, and additional sections similar to figure 6.

*Author contributions.* AB and AR conceived the study, AB implemented the isochronal tracer scheme, conducted the analysis, and wrote the paper with input from AR. AR generated the climate forcing, implemented the basal friction optimization, and designed the ensemble with input from AB.

*Competing interests.* The authors declare no competing interests.

*Acknowledgements.* AB acknowledges support from the Trond Mohn Foundation. Alexander Robinson was supported by the Ramón y Cajal Programme of the Spanish Ministry for Science, Innovation and Universities (grant no. RYC-2016-20587), as well as the Spanish Ministry of Science and Innovation project ICEAGE (grant no. PID2019-110714RA-I00).



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
