# Peer review of "Modeling the Greenland englacial stratigraphy"

_The Cryosphere, 2020_

## Referee Comment (RC1) · Nicholas Holschuh (Referee) · 16 Jan 2021

**Review of:** Modeling the Greenland englacial stratigraphy
**Submitted to:** The Cryosphere Discussions
**Reviewer:** Nicholas Holschuh

**General Comments:**

In previous work, Born derived and demonstrated a model of ice flow focused on the evolution of layer packages (Born, 2017). An explicit output from this model is the isochronal layer geometry – a field that is directly comparable with radar observations, which have the potential to provide a spatially and temporally comprehensive check on model performance. The formulation presented in (Born, 2017) outperformed existing models which use Eulerian velocity fields to contour the age field of the ice sheet, avoiding issues of numerical diffusion that can result in unrealistically smooth age fields. That work forms the backbone of this manuscript, which is focused on making that framework modular, such that it can be applied to existing 3D models of ice flow and allow for the use of observed englacial layers in model tuning.

At its core, this is a methodological paper, pursuing an important objective at the cutting edge of ice sheet modeling. But the authors spend most of the paper discussing the specifics of their model *results* – what drives model-layer / observed-layer mismatch and the interaction between specific tuning parameters in Yelmo (the underlying ice physics engine (Robinson et al., 2020)). This would be important if the tuned model (i.e. the depth-age model of Greenland since LGM) were the central product of this work, but the real scientific contribution here is what the authors have learned about the *process* – that (1) it is possible to apply the layer tracking scheme in (Born, 2017) to 3D models that do not explicitly track layers, that (2) the resulting layers are an improvement over results of previous methods, that (3) tuning ice flow models (or at least, this ice flow model) to the ice thickness alone can result in large errors in englacial dynamics, and (4) that it is *important* that future models use this method, as Eulerian tracers produce a systematic bias in model-layer age.

While I have only minor questions about the technical work done, the changes that I think are most necessary are to the writing, to maximize the paper's impact and ensure that the scientific contribution of the work is clear. Right now, the key messages are buried in extensive description of Greenland accumulation, and the large, multi-panel figures of model mismatch do little to articulate this work's core message. In the technical comments below, I provide specific changes that I think will help resolve these issues. Ultimately, the layer tracking module developed here has the potential to be a widely used tool and help constrain models across a wide range of complexities, and I want to ensure this work has the impact it deserves.

**Technical Comments:**

If this were simply another model of Greenland from LGM to today, the scientific contribution would be limited, as there is no articulated "experiment" here probing Greenland dynamics. The discussion of errors in model forcing provides insight into the climate parameterizations chosen, but they distract from the methodological improvements that will be this paper's legacy. To make clear the scientific contribution, I think three primary changes are required:

1. There should be a reproducible description of how the layer tracing scheme couples to the 3D model. There is extensive description of the climate spin up, and the model parameters being tuned, but no description of the implementation that translates output from (Robinson et al., 2020) to input in (Born 2017). This should (1) make clear to the reader exactly how this method avoids the pitfalls of Eulerian tracers, especially while using an ice sheet model that solves the physical equations on an Eulerian depth grid, and (2) enable future application of the method to other ice sheet models.

2. A more succinct description of the optimal model should be provided, but primarily to highlight which model parameters are sensitive to the stratigraphic constraint (indicating which processes / boundary forcings this optimization approach is likely to capture). The extensive description of figures 4, 5, 10, and 11 can be substantially trimmed. In addition, I think the readers would benefit from a deeper explanation of the differences observed and the drivers of that difference in figure 12, which demonstrates the value of the improved parameter optimization.

3. At present, this paper avoids the discussion of an important and active area of research: fitting layer shapes in the dynamic regions of Greenland and Antarctica. This is in-part because the outlet glacier modeling done here is simplified. But layer fitting in these areas has the potential to capture spatial and temporal heterogeneity in the basal boundary condition that no other method can address, and given the high-profile nature of features like the layer draw-down in Northeast Greenland (e.g., Fahnestock et al., 2001) and the complex folding at Petermann Glacier (e.g., Bons et al., 2016), I think it would be appropriate for this work (especially given its title: "Modeling the Greenland englacial stratigraphy") to directly address dynamically controlled folding which dominates the marginal ice. There is an extensive literature on the Weertman Effect (e.g., Hindmarsh et al., 2006; Leysinger Vieli et al., 2007; Parrenin et al, 2007; Wolovick et al., 2016), models of layer shape in Greenland (Leysinger Vieli et al., 2018), and direct comparison of models and data (e.g., Holschuh et al., 2019) that could be used to substantiate the need and interest in developing these layer modeling methods. While you could never provide a complete review of the literature here, the repeated claim that accumulation history is the dominant variable relies on an implicit assumption that we ignore the dynamic outlet glaciers. A full description of your method should include its applicability to the ice sheet margins and how it fits within the existing literature on the subject.

These comments are motivated by the fact that I would like to see this method applied more broadly! A revision focused on clarity of the method, quantification of its improvement over previous methods, and guidelines for its future use will make the scientific merits and novelty clear.

**Line-Item Corrections:**

| | |
|---|---|
| Page #: 1
Line #: 6-8 | I find this description confusing, as mass transfer happens within Yelmo (outside of the layer evolution scheme). How is it that mass transfer between layers is avoided when the solver exists in depth, not time? (I think this is just part of a larger desire to see a clear description of the coupling). |
| Page #: 1
Line #: 10 | The phrasing "… selecting simulations…" is unclear here -- selecting them for what? Perhaps rephrase to "Using an ensemble of simulations to optimize climate and ice dynamic parameter selection, we show that direct comparison with the dated radiostratigraphy data yields notably more accurate results than choosing parameters based on fit to total ice thickness alone." |
| Page #: 1
Line #: 16-22 | I appreciate the oceanographic analogy here, it adds nice context! |
| Page #: 1
Line #: 21 | "The proverbially glacial flow" -- I'm not sure what you mean here by "proverbially". |
| Page #: 2
Line #: 28-31 | There is an error in construction here, with the sentence that reads: "The ... layers could aid…, where to find…, to reconstruct…, or to determine…." Either each clause should start with an infinitive, or they should follow from a common verb. It could be "The layers could aid, find, reconstruct, or determine", or it could be "The layers could help us to select, to find, to reconstruct, or to determine". |
| Page #: 2
Line #: 36 | Should be "finite-difference" not "finite-differences" |
| Page #: 3
Line #: 63-64 | Given that your layer thicknesses are smaller than the vertical resolution of the solver, I am still a bit confused about how you can solve for changes in layer thickness and still guarantee no numerical diffusion? (This is where a discussion of the coupling would be helpful). |
| Page #: 0
Line #:
Section 3.2 | I had difficult following the narrative through this section, especially Figure 10 and 11. If you intend to keep all of this, it would help to guide the reader through it in a more directed way -- referring to specific panels in the figures (not just a grid of 40 Greenlands), and pointing back to the motivating questions that justify the extensive description provided. Ultimately, I did not see any need for detailed description of the specific model output you provided, as there are more sophisticated modeling exercises that one could turn to for full description of dynamics in Greenland. But if you think there is value in dissecting the specifics of this model configuration and output (as opposed to just focusing on the exercise of modeling and optimizing), you need to motivate that more clearly somewhere. |

| | |
|---|---|
| Page #: 19
Line #: 315-
329 | This section had me thinking about a more general question -- do isochrones add value in constraining processes outside of the time range that they span? Making an explicit statement about how temporal coverage of the data impacts temporal constraint in the model could be very interesting. |
| Page #: 19
Line #: 345 | Space between numbers and units. |
| Page #: 20
Line #: 350 | You regularly state that the Eulerian age tracer (orange curve) in Figure 12 shows older ages, but in all situations it seems that the age of the orange curve falls below the blue curve. Am I misreading Figure 12? "Older" continually appears in your description of the Eulerian method, and I am having trouble rectifying that with the figure. |
| Page #: 20
Line #: 351-
353 | A clear description of the coupling will certainly answer this question, but somewhere depth and age must be mapped to one another to couple the ice flow model to the layer evolution model, and I'm still not clear on how the horizontal flow speeds within a given layer are calculated (to prevent flow across boundaries). |
| Page #: 20
Line #: 357-
358 | Okay, I think I understand the "older" comment here -- if a model were optimized using the Eulerian scheme, the true model age (when calculated correctly using the new layer evolution scheme) would actually be older than the constraint. But that seems different than the previous statement, that the Eulerian tracer data produces older ages. I am probably just confused, but some clarity through this whole section on the nature of the bias of the Eulerian method would be useful. |
| Page #: 21
Line #: 378-
380 | I think this sentence is a bit of a tautology -- the model calibrated to the stratigraphic data fits the stratigraphic data better. It would be better to appeal to a third target variable to evaluate accuracy. Something like: "Models that are optimized to match the ice thickness require unrealistic precipitation histories, resulting in erroneous layer ages. These precipitation histories can be ruled out when constraining model parameters with both thickness and layer age." |
| Page #: 21
Line #: 381-
386 | This paragraph is only true because you exclude the dynamic regions of Greenland from your analysis. Of course accumulation matters more when dynamic vertical velocities are otherwise very small. Where the Weertman effect is large, surface mass balance history will be much less important. This is why I advocate for a broader discussion of what affects layer shapes near the margins, contextualized in the literature. |

| Page #: 23 | Again, this point that "accumulation eclipses the impact of dynamics" is not |
| Line #: 423 | universally true, and will depend on the target region of interest for future applications |
| | of this method. In the outlet glaciers, this is far less likely to be the case, so I would |
| | hesitate to use such strong language when this method could be applied beyond just |
| | the interior of ice sheets as was done here. |

**References:**

Bons, P.D., Jansen, D., Mundel, F., Bauer, C.C., Binder, T., Eisen, O., Jessell, M.W., Llorens, M.-G., Steinbach, F., Steinhage, D., and Weikusat, I., 2016, Converging flow and anisotropy cause large-scale folding in Greenland's ice sheet.: Nat. Commun., v. 7, p. 11427, doi: 10.1038/ncomms11427.

Fahnestock, M., Abdalati, W., Joughin, I., Brozena, J., and Gogineni, P., 2001, High geothermal heat flow, Basal melt, and the origin of rapid ice flow in central Greenland.: Science, v. 294, p. 2338–2342, doi: 10.1126/science.1065370.

Hindmarsh, R.C.A., Leysinger Vieli, G.J.M.C., Raymond, M.J., and Gudmundsson, G.H., 2006, Draping or overriding: The effect of horizontal stress gradients on internal layer architecture in ice sheets: Journal of Geophysical Research: Earth Surface, v. 111, doi: 10.1029/2005JF000309.

Holschuh, N., Lilien, D., and Christianson, K., 2019, Thermal Weakening, Convergent Flow, and Vertical Heat Transport in the Northeast Greenland Ice Stream Shear Margins: Geophysical Research Letters, v. 46, p. 8184–8193.

Leysinger Vieli, G.J.-M.C., Martín, C., Hindmarsh, R.C.A., and Lüthi, M.P., 2018, Basal freeze-on generates complex ice-sheet stratigraphy: Nature Communications, v. 9, p. 1–13, doi: 10.1038/s41467-018-07083-3.

Leysinger Vieli, G.J.-M.C., Hindmarsh, R.C. a, and Siegert, M.J., 2007, Three-dimensional flow influences on radar layer stratigraphy: Annals of Glaciology, p. 22–28.

Parrenin, F., and Hindmarsh, R., 2007, Influence of a non-uniform velocity field on isochrone geometry along a steady flowline of an ice sheet: Journal of Glaciology, v. 53, p. 612–622, doi: 10.3189/002214307784409298.

Wolovick, M.J., and Creyts, T.T., 2016, Overturned folds in ice sheets: Insights from a kinematic model of traveling sticky patches and comparisons with observations: Journal of Geophysical Research: Earth Surface, v. 121, p. 1065–1083, doi: 10.1002/2015JF003698

---

## Referee Comment (RC2) · Anonymous Referee #2 · 11 Feb 2021

This model presents a modeling of the Greenland ice sheet comprising a module for solving the age equation, which allows to compare the modeled isochrones with radar-observed isochrones. This age module is derived from the study of Born (2017), but it is now decoupled from the thermo-mechanical model used. The work of Born (2017) demonstrated how this new numerical scheme, based on the time domain, outperforms classical Eulerian schemes which show large diffusive artifacts.

This study presents itself as "the first three-dimensional ice sheet model that explicitly simulates the Greenland englacial stratigraphy". The ice sheet model used here is YELMO. The climate forcing used is based on two snapshots (Present-day and LGM) and a climate index based on paleoclimatic archives. A pseudo inverse method is used to fit a few parameters so that the model best fits either the present-day topography, the

internal layering, or both. It is found that the model fitted onto the englacial stratigraphy gives a better overall fit than the model fitted onto the surface topography. So it is "easy" to have a model that fit the surface topography for wrong reasons.

The article is generally clearly written, and is a good contribution to the field of ice sheet modeling and comparison to observations. It certainly opens a new chapter of model-data comparison in ice sheet modeling.

My main comments are similar to the remarks of the first reviewer: I reckon that some sections are too detailed while others are not enough, so that the main output is too diluted to be easily accessible. I feel that the method section of the article describe in too much details the YELMO ice sheet model used and its climate forcing, instead of focusing on the new age numerical scheme and how the variable are transferred from the thermo-mechanical grid to the age grid. Then, the results section describe in details the fit of the model to the observations, while the inverse method used here is really basic and does not allow to explore the full parameter space. I would rather focus on the comparison of this new age numerical scheme with previous numerical schemes, like the Eulerian and semi-Lagrangian schemes. In short, I reckon the value of this article is more on the age numerical scheme than on the inverse method.
* * *

---

## Author Comment (AC1) · 26 Feb 2021

We thank reviewer Holschuh for the thorough evaluation and detailed suggestions. We agree with the characterization of our manuscript as more of a methodological paper and that this calls for a thorough technical description. This is a very useful comment, because we originally understood the previous publication (Born, 2017) as the primary technical reference and the present manuscript as a first application. However, it is true that the coupling between the layer tracing scheme and the physical host model is not trivial and warrants a more detailed and reproducible description. Related to that, we also plan to make our source code publicly available upon acceptance.

Seeing that the comments on the technical work are minor, we plan to revise our

manuscript following the three suggested primary changes:

1) A reproducible description of the coupling with the host model in Section 2.4. In addition to extending the written explanation, we propose adding a flowchart to illustrate how and when information flows.

2) A more succinct description of the results. This means that in addition to extending Section 2.4, we will reduce the descriptions of figures 4, 5, 10, 11 where possible.

3) A discussion of how our work relates to ongoing studies of fitting isochronal surfaces in dynamically active regions. We also agree that the manuscript would greatly benefit from a discussion of recent efforts to use isochronal layers as constraints for dynamical processes near the margins, although this is not the focus of our work. In addition to the dynamic shortcomings of our coarse-resolution model in these regions, the isochronal grid does currently not allow for folding, i.e., a non-monotonous increase of age with depth. This will be clarified in the revised methods section. We are working toward the application of the layer tracing scheme to dynamically active regions, but this will require further development work to improve, e.g., the numerical efficiency. This information will be included in the revised text.

---

## Author Comment (AC2) · 26 Feb 2021

We would like to thank reviewer #2 for an encouraging review. The primary criticism echoes that of reviewer #1, to provide more detail on the isochronal layer tracing scheme and how variables are transferred between it and the host model. We agree with this criticism and will address it in the revised manuscript. The goal is to have a reproducible description that will serve as a technical reference for future papers. We plan to add technical details of the coupling with the host model in Section 2.4 and propose adding a flowchart to illustrate how and when information flows.

---

## Author Response (AR1)

We would like to thank both reviewers for their careful work and encouraging comments. We reply to all comments below. Revised text in the manuscript is highlighted with red.

**Reviewer #1**

General Comments:
In previous work, Born derived and demonstrated a model of ice flow focused on the evolution of layer packages (Born, 2017). An explicit output from this model is the isochronal layer geometry – a field that is directly comparable with radar observations, which have the potential to provide a spatially and temporally comprehensive check on model performance. The formulation presented in (Born, 2017) outperformed existing models which use Eulerian velocity fields to contour the age field of the ice sheet, avoiding issues of numerical diffusion that can result in unrealistically smooth age fields. That work forms the backbone of this manuscript, which is focused on making that framework modular, such that it can be applied to existing 3D models of ice flow and allow for the use of observed englacial layers in model tuning.

At its core, this is a methodological paper, pursuing an important objective at the cutting edge of ice sheet modeling. But the authors spend most of the paper discussing the specifics of their model results – what drives model-layer/observed-layer mismatch and the interaction between specific tuning parameters in Yelmo (the underlying ice physics engine (Robinson et al., 2020)). This would be important if the tuned model (i.e. the depth-age model of Greenland since LGM) were the central product of this work, but the real scientific contribution here is what the authors have learned about the process – that (1) it is possible to apply the layer tracking scheme in (Born, 2017) to 3D models that do not explicitly track layers, that (2) the resulting layers are an improvement over results of previous methods, that (3) tuning ice flow models (or at least, this ice flow model) to the ice thickness alone can result in large errors in englacial dynamics, and (4) that it is important that future models use this method, as Eulerian tracers produce a systematic bias in model-layer age.

While I have only minor questions about the technical work done, the changes that I think are most necessary are to the writing, to maximize the paper's impact and ensure that the scientific contribution of the work is clear. Right now, the key messages are buried in extensive description of Greenland accumulation, and the large, multi-panel figures of model mismatch do little to articulate this work's core message. In the technical comments below, I provide specific changes that I think will help resolve these issues. Ultimately, the layer tracking module developed here has the potential to be a widely used tool and help constrain models across a wide range of complexities, and I want to ensure this work has the impact it deserves.

If this were simply another model of Greenland from LGM to today, the scientific contribution would be limited, as there is no articulated "experiment" here probing Greenland dynamics. The discussion of errors in model forcing provides insight into the climate parameterizations chosen, but they distract from the methodological improvements that will be this paper's legacy. To make clear the scientific contribution, I think three primary changes are required:

We are grateful for the detailed and constructive criticism of this review. We agree with the comments and implemented almost all of the suggested changes.

1. There should be a reproducible description of how the layer tracing scheme couples to the 3D model. There is extensive description of the climate spin up, and the model parameters being tuned, but no description of the implementation that translates output from (Robinson et al., 2020) to input in (Born 2017). This should (1) make clear to the reader exactly how this method avoids the pitfalls of Eulerian tracers, especially while using an ice sheet model that solves the physical equations on an Eulerian depth grid, and (2) enable future application of the method to other ice sheet models.

We extended section 2.4 with additional text and one additional figure that illustrates the algorithm during one time step, what input data it requires from the host model, and when.

2. A more succinct description of the optimal model should be provided, but primarily to highlight which model parameters are sensitive to the stratigraphic constraint (indicating which processes / boundary forcings this optimization approach is likely to capture). The extensive description of figures 4, 5, 10, and 11 can be substantially trimmed. In addition, I think the readers would benefit from a deeper explanation of the differences observed and the drivers of that difference in figure 12, which demonstrates the value of the improved parameter optimization.

We agree that the description of the composite analysis was too long. This text and the former figures 10 and 11 have now been moved to an appendix. The main findings are summarized in section 3.2 as a single paragraph. We believe that the discussion of former figures 4 and 5 is already quite compact and important as an introduction to how the simulated isochrones can be compared with the reconstructed stratigraphy. We think this is important to build intuition before discussing the RMSE of the ensemble simulations and therefore prefer to keep section 3.1 unchanged.
At present, we cannot provide additional detail on the sensitivity of individual model parameters. We were surprised by the apparently weak sensitivity of the simulated isochrones to dynamic model parameters, which appears to contradict findings from Born (2017). We suspect the optimization of basal friction to have a strong influence on this result, which is part of the learning process that Reviewer #1 refers to above. This explanation is now included at multiple locations in the manuscript and we plan to revisit the issue in a future study.
Lastly, the description of former figure 12 (now 11) has been revised to make it clearer. Although that did not make the text much longer, we hope that the now more precise description more clearly conveys the added value of our new layer advection scheme. See also our reply to the specific questions below.

3. At present, this paper avoids the discussion of an important and active area of research: fitting layer shapes in the dynamic regions of Greenland and Antarctica. This is in-part because the outlet glacier modeling done here is simplified. But layer fitting in these areas has the potential to capture spatial and temporal heterogeneity in the basal boundary condition that no other method can address, and given the high-profile nature of features like the layer draw-down in Northeast Greenland (e.g., Fahnestock et al., 2001) and the complex folding at Petermann Glacier (e.g., Bons et al., 2016), I think it would be appropriate for this work (especially given its title: "Modeling the Greenland englacial stratigraphy") to directly address dynamically controlled folding which dominates the marginal ice. There is an extensive literature on the Weertman Effect (e.g., Hindmarsh et al., 2006; Leysinger Vieli et al., 2007; Parrenin et al, 2007; Wolovick et al., 2016), models of layer shape in Greenland (Leysinger Vieli et al., 2018), and direct comparison of models

and data (e.g., Holschuh et al., 2019) that could be used to substantiate the need and interest in developing these layer modeling methods. While you could never provide a complete review of the literature here, the repeated claim that accumulation history is the dominant variable relies on an implicit assumption that we ignore the dynamic outlet glaciers. A full description of your method should include its applicability to the ice sheet margins and how it fits within the existing literature on the subject.

We revised the last section to discuss how we can constrain the ice dynamics and limitations of the current method. This discussion includes references to most of the studies mentioned above. Note that in its present form, our isochrone advection scheme uses a vertical axis where age has to increase monotonically. This does not allow for layer folding.

Line-Item Corrections:
Page #: 1
Line #: 6-8
I find this description confusing, as mass transfer happens within Yelmo (outside of the layer evolution scheme). How is it that mass transfer between layers is avoided when the solver exists in depth, not time? (I think this is just part of a larger desire to see a clear description of the coupling).

We rephrased this part of the abstract slightly and hope that the revised description in section 2.4 clarifies this question.

Page #: 1
Line #: 10
The phrasing "... selecting simulations..." is unclear here -- selecting them for what? Perhaps rephrase to "Using an ensemble of simulations to optimize climate and ice dynamic parameter selection, we show that direct comparison with the dated radiostratigraphy data yields notably more accurate results than choosing parameters based on fit to total ice thickness alone."

We have changed the verb to "calibrating".

Page #: 1
Line #: 16-22
I appreciate the oceanographic analogy here, it adds nice context!

Thank you.

Page #: 1
Line #: 21
"The proverbially glacial flow" -- I'm not sure what you mean here by "proverbially".

This is a failed attempt at a play of words. We were trying to emphasize the relative slowness of ice sheet dynamics with the figurative sense of the word "glacial", while naturally also referring to "glacial flow" in the literal sense. The word "proverbial" has now been removed from the text.

Page #: 2

Line #: 28-31
There is an error in construction here, with the sentence that reads: "The ... layers could aid..., where to find..., to reconstruct..., or to determine...." Either each clause should start with an infinitive, or they should follow from a common verb. It could be "The layers could aid, find, reconstruct, or determine", or it could be "The layers could help us to select, to find, to reconstruct, or to determine".

This has been corrected.

Page #: 2
Line #: 36
Should be "finite-difference" not "finite-differences"

corrected

Page #: 3
Line #: 63-64
Given that your layer thicknesses are smaller than the vertical resolution of the solver, I am still a bit confused about how you can solve for changes in layer thickness and still guarantee no numerical diffusion? (This is where a discussion of the coupling would be helpful).

Please see section 2.4.

Page #: 0
Line #:
Section 3.2
I had difficult following the narrative through this section, especially Figure 10 and 11. If you intend to keep all of this, it would help to guide the reader through it in a more directed way -- referring to specific panels in the figures (not just a grid of 40 Greenlands), and pointing back to the motivating questions that justify the extensive description provided. Ultimately, I did not see any need for detailed description of the specific model output you provided, as there are more sophisticated modeling exercises that one could turn to for full description of dynamics in Greenland. But if you think there is value in dissecting the specifics of this model configuration and output (as opposed to just focusing on the exercise of modeling and optimizing), you need to motivate that more clearly somewhere.

We agree that 2 * 40 Greenlands is too much for the main body of the paper but would like to keep this figure as a reference for interested readers and therefore suggest moving it to the appendix. We think that a broad discussion of physical properties like isochrone depth is useful in addition to the RMSE and so we include the most important findings from the full discussion as a summary at the end of section 3.2.

Page #: 19
Line #: 315-329
This section had me thinking about a more general question -- do isochrones add value in constraining processes outside of the time range that they span? Making an explicit statement about how temporal coverage of the data impacts temporal constraint in the model could be very interesting.

Yes they do. Additional precipitation during the glacial period leads to a higher ice sheet and steeper surface gradients. Additional accumulation that falls on top will therefore experience a faster advection and hence more dynamic thinning. This is clearly seen in the reduced depth of the 11.7 ka isochrone when comparing the high $f_{LGM}$ composite with the ensemble average (Fig. A2).
We included this explanation in the greatly shortened discussion of the ensemble composites in the main text.

Page #: 19
Line #: 345
Space between numbers and units.

corrected

Page #: 20
Line #: 350
You regularly state that the Eulerian age tracer (orange curve) in Figure 12 shows older ages, but in all situations it seems that the age of the orange curve falls below the blue curve. Am I misreading Figure 12? "Older" continually appears in your description of the Eulerian method, and I am having trouble rectifying that with the figure.

This section was not as clear as it could be and we made several changes to the text to improve it. There are two "Eulerian curves in figure 11 (new numbering)". The dashed blue curve shows the Eulerian tracer diagnostic for the $BEST_{all}$ simulation, i.e., the Eulerian diagnostic for a simulation calibrated with the isochronal scheme. It can be used to directly compare the quality of the Eulerian age tracer with our new scheme (blue solid). Here, ages are older because of upward numerical diffusion. In addition, the orange curve shows another simulation that was calibrated using the Eulerian age tracer, but shows the age profile as simulated by the more reliable isochrone scheme. Again, this data can be compared directly with the solid blue curve.

Page #: 20
Line #: 351-353
A clear description of the coupling will certainly answer this question, but somewhere depth and age must be mapped to one another to couple the ice flow model to the layer evolution model, and I'm still not clear on how the horizontal flow speeds within a given layer are calculated (to prevent flow across boundaries).

We hope that the revised section 2.4 answers this question. The horizontal flow speeds within a given isochronal layer are interpolated from the coarser Yelmo grid. This means that they are calculated in the Eulerian host model and subject to numerical diffusion. However, the velocity field varies rather smoothly with depth and the main variable that controls ice viscosity, temperature, is strongly influenced by *physical* diffusion that significantly lessens the impact of its spurious numerical counterpart.

Page #: 20
Line #: 357-358
Okay, I think I understand the "older" comment here -- if a model were optimized

using the Eulerian scheme, the true model age (when calculated correctly using the new layer evolution scheme) would actually be older than the constraint. But that seems different than the previous statement, that the Eulerian tracer data produces older ages. I am probably just confused, but some clarity through this whole section on the nature of the bias of the Eulerian method would be useful.

This is correct. All simulations simulate both the new layer scheme and the Eulerian age tracer. We only show the latter once, as the dashed blue line in figure 11, and we discuss a single simulation that was calibrated using the Eulerian tracer, shown as orange curves in figures 9, 11, and 12. Unlike BEST$_{all}$ and BEST$_{ice}$, this simulation does not have a specific name.

Page #: 21
Line #: 378-380
I think this sentence is a bit of a tautology -- the model calibrated to the stratigraphic data fits the stratigraphic data better. It would be better to appeal to a third target variable to evaluate accuracy. Something like: "Models that are optimized to match the ice thickness require unrealistic precipitation histories, resulting in erroneous layer ages. These precipitation histories can be ruled out when constraining model parameters with both thickness and layer age."

We believe that the original text was accurate and not stating an obvious fact because it referred to the finding that the simulation calibrated with the stratigraphic constraint also fits the *ice thickness* better. However, we appreciate the suggestion to rephrase the text and would like to include an only slightly modified version.

Page #: 21
Line #: 381-386
This paragraph is only true because you exclude the dynamic regions of Greenland from your analysis. Of course accumulation matters more when dynamic vertical velocities are otherwise very small. Where the Weertman effect is large, surface mass balance history will be much less important. This is why I advocate for a broader discussion of what affects layer shapes near the margins, contextualized in the literature.

We modified the first and last sentences of this paragraph and revised the part of the discussion that follows it. Please see our reply to general comment #3 above.

Page #: 23
Line #: 423
Again, this point that "accumulation eclipses the impact of dynamics" is not universally true, and will depend on the target region of interest for future applications of this method. In the outlet glaciers, this is far less likely to be the case, so I would hesitate to use such strong language when this method could be applied beyond just the interior of ice sheets as was done here.

We agree that this statement was too strong and modified it accordingly.

**Reviewer #2**

This model presents a modeling of the Greenland ice sheet comprising a module for solving the age equation, which allows to compare the modeled isochrones with radar-observed isochrones. This age module is derived from the study of Born (2017), but it is now decoupled from the thermo-mechanical model used. The work of Born (2017) demonstrated how this new numerical scheme, based on the time domain, outperforms classical Eulerian schemes which show large diffusive artifacts.

This study presents itself as "the first three-dimensional ice sheet model that explicitly simulates the Greenland englacial stratigraphy". The ice sheet model used here is YELMO. The climate forcing used is based on two snapshots (Present-day and LGM) and a climate index based on paleoclimatic archives. A pseudo inverse method is used to fit a few parameters so that the model best fits either the present-day topography, the internal layering, or both. It is found that the model fitted onto the englacial stratigraphy gives a better overall fit than the model fitted onto the surface topography. So it is "easy" to have a model that fit the surface topography for wrong reasons.

The article is generally clearly written, and is a good contribution to the field of ice sheet modeling and comparison to observations. It certainly opens a new chapter of model-data comparison in ice sheet modeling.

My main comments are similar to the remarks of the first reviewer: I reckon that some sections are too detailed while others are not enough, so that the main output is too diluted to be easily accessible. I feel that the method section of the article describe in too much details the YELMO ice sheet model used and its climate forcing, instead of focusing on the new age numerical scheme and how the variable are transferred from the thermo-mechanical grid to the age grid. Then, the results section describe in details the fit of the model to the observations, while the inverse method used here is really basic and does not allow to explore the full parameter space. I would rather focus on the comparison of this new age numerical scheme with previous numerical schemes, like the Eulerian and semi-Lagrangian schemes. In short, I reckon the value of this article is more on the age numerical scheme than on the inverse method.

We thank Reviewer #2 for this evaluation of our work. We agree that the primary contribution of this study is the method itself rather than the results that we were able to obtain from it so far. It was our intent to clearly outline what we have learned to this point and how future work may improve upon ours. In our understanding, Reviewer #2 raises two major points: 1) A lack of detail in the description of the isochronal tracer advection scheme, in particular in comparison to the detailed description of the host model Yelmo, and 2) an incomplete description of how our methodology is superior to alternative numerical schemes.

We addressed the first point by extending the technical description in section 2.4, including a new figure showing the execution flow during one time step. We would like to keep the description of Yelmo in its current form, because the model is relatively new and many of the technical details have not yet been published.

With regard to the second point, we revised section 3.3 and the comparison with the Eulerian age tracer. The comments of Reviewer #1 made us realize that this part was not

particularly well written and important points likely to be misunderstood. We chose not to extend the comparison with the semi-Lagrangian scheme, because we cannot assess it directly and therefore cannot say much more than what is already included in the discussion: " Compared to Lagrangian or semi-Lagrangian tracer advection schemes, our method avoids costly interpolation or low particle densities. It is possible to use an uneven spacing of the isochronal grid to concentrate computational cost to key periods of interest."

---

## Referee Report (RR1)

**Review of:** Modeling the Greenland englacial stratigraphy
**Submitted to:** The Cryosphere
**Reviewer:** Nicholas Holschuh

**General Comments:**

This work focuses on the implementation of an isochrone dating method for 3D ice sheet models, which solves for the evolving thickness of englacial layers (and therefore the depth-age scale) avoiding the numerical diffusion of existing Eulerian and Lagrangian tracer methods for constraining layer ages. After the revisions, I have two minor points requiring clarification (discussed below), but the reorganization of this manuscript together with the added text and figures makes the message very clear. The authors do a great job of explaining the coupling between the ice sheet model and the layer following module, such that it can be easily implemented by other modelers going forward. I would be happy to recommend it for publication once the points below are clarified.

**Technical Comments:**

My two remaining questions are, to a certain extent, updates to questions I asked in the first review. The first point is primarily a clarification of what seems like a discrepancy in the figures, while the second is a suggestion to remove any ambiguity in the discussion of model bias:

1. Figures 5, 6, and 7 – I am having difficulty rectifying the cross-section you've provided [F7] with the map differences presented earlier [F5-6]. Using your cross-section for the 11.7 ka layer, I find that the BEST_all Δdepth should have values of about ~250 m through the full central part of Greenland, but the map appears to have values of ~100 m at most? Could you verify that those values are being plotted the way you expect? I also think it would be helpful if you could expand the Δdepth color scale so that Figure 5 is not entirely saturated – the point that it is uniformly positive could still be made while illustrating the magnitude of the deviation (which, right now, can only be inferred to be >= 320 m).

2. Figure 11 and line 332 – I understand the distinction you've made in the response to the previous review regarding "younger" vs "older" when describing bias, but I still stumble quite a bit with the language describing the orange curve in Figure 11. Here, you state there is an old bias (line 332). It would be helpful to be explicit about what specifically you are describing as biased (the Eulerian derived ages or the true model) and what they are biased relative to (the observed ages at those sites, the true ages of the model selected as optimal using the layer following scheme, or the true ages of the model selected based on the optimal Eulerian derived ages). By the time I get to this sentence, I am already convinced by the previous paragraph that the Eulerian method calculates inaccurately old ages, so I don't think it adds value to say here that the Eulerian method is biased relative to the layer following method. Instead, the new piece of information is in the resulting ice sheet model bias. If you optimize your model to match the Eulerian ages to the observed ages, your model will actually produce an ice sheet that is *younger* than the true ice sheet (because the Eulerian ages that you've fit to are older than the true ages of the ice sheet, as shown by the layer following scheme). Phrasing things this way has the advantage that it is immediately apparent from Figure 11 -- the orange curve falls to the left of both the solid blue curve and the black curve.

Beyond these two points of clarification, I only have a few line-item comments. I really enjoyed the paper!

**Line-Item Corrections:**

| | |
|---|---|
| Page #: 2
Line #: 55 | I'm not sure what you mean here by "spatial patterns that differ on the various isochronal surfaces." Patterns of what? Do you mean spatial patterns for isochrone depth error? It would help to be specific here. |
| Page #: 3
Line #: 72 | It might be worth mentioning that N is the effective pressure assuming no support from a pressurized subglacial hydrologic system, just to be clear that effective pressure could have another term that is omitted here. |
| Page #: 10
Line #: Figure 4 | I think this is a great addition! I had one clarifying question -- the vertical interpolation you refer to here is for the Yelmo velocities, right? If so, should that either be in blue, or be the step following the input of the velocities from Yelmo? It might be helpful to state what is being interpolated and what is being advected in those two boxes, just to be totally explicit. |
| Page #: 11
Line #: 229 | You're missing the section reference here -- I think it should read: "… by discussing two examples from the ensemble in section 3.1, followed by…" |
| Page #: 12
Line #: Figure 5 | As mentioned above, I think it would be helpful to expand the color scale here so that the interior values are not saturated (to get a sense for the full magnitude of the error). |
| Page #: 13
Line #: Figure 6 | The panel showing error for the 11.7 ka error seems to have values with a maximum between 80 and 160 m in the Greenland interior, but that is quite different from the ~250 m errors shown in Figure 7. Is there something I'm missing here? |
| Page #: 17
Line #: 301 | The word "perspective" here is vague -- I think it might be better to rephrase to something more specific, for example: "… the high resolution of our model in the temporal domain allows for a different and complementary analysis, using the full age profile at certain locations to constrain model performance." |
| Page #: 17
Line #: 317 | To assist the reader and maintain parallel structure here, it would be useful to point out that the Eulerian tracer is the dashed blue line in Fig. 11 and the isochronal tracing scheme is solid blue. The sentence as written references both methods but only one line in the figure, which can be confusing. IE: "Comparison between the Eulerian age tracer (dashed blue line) and the ioschronal age tracer (solid blue line) applied to the BEST_all simulation shows clear disagreement between the two methods for defining the depth-age scale." |

Page #: 17
Line #: 325-332

This is the section that I got a bit hung up on. I apologize if I said the opposite of this in my previous review (I think I might have…), but I realize now this sentence would be clearer if you talked about it in terms of parameter selection instead of model calibration. I tried my hand at rephrasing to clarify the points that confused me, with the hope that my attempted rephrasing will help you see where my issues were: "The spurious bias toward older ice with the Eulerian scheme has a noticeable effect on model behavior when parameter optimization is based on its output ages (as in the model plotted in orange, Fig. 11). Here we show the ages generated using the isochronal scheme, derived from model output from an ice sheet simulation using parameters chosen to optimize the quality of fit for isochrones generated using the Eulerian age tracer. Note that this is a different simulation from the one plotted in blue, which uses parameters chosen based on quality of fit for isochrones generated using the isochronal scheme."

Then, in line 332, I think the salient point is that models which optimize their boundary conditions using Eulerian derived ages will produce ice sheets with true ages that are younger than observations (as shown in Figure 11). This makes a clear distinction with the previous paragraph. There, you show that the Eulerian method produces older ages than the isochronal method -- here you show that model optimization based on Eulerian ages biases the ice sheet toward younger ice deeper in the column.

---

## Author Response (AR2)

We thank reviewer Nicholas Holschuh again for his very detailed and helpful comments. We reply to all comments below. Revised text in the manuscript is highlighted with red.

**Reviewer #1, Nicholas Holschuh**

**General Comments:**
This work focuses on the implementation of an isochrone dating method for 3D ice sheet models, which solves for the evolving thickness of englacial layers (and therefore the depth-age scale) avoiding the numerical diffusion of existing Eulerian and Lagrangian tracer methods for constraining layer ages. After the revisions, I have two minor points requiring clarification (discussed below), but the reorganization of this manuscript together with the added text and figures makes the message very clear. The authors do a great job of explaining the coupling between the ice sheet model and the layer following module, such that it can be easily implemented by other modelers going forward. I would be happy to recommend it for publication once the points below are clarified.

**Technical Comments:**
My two remaining questions are, to a certain extent, updates to questions I asked in the first review. The first point is primarily a clarification of what seems like a discrepancy in the figures, while the second is a suggestion to remove any ambiguity in the discussion of model bias:

1. Figures 5, 6, and 7 – I am having difficulty rectifying the cross-section you've provided [F7] with the map differences presented earlier [F5-6]. Using your cross-section for the 11.7 ka layer, I find that the BEST_all Δdepth should have values of about ~250 m through the full central part of Greenland, but the map appears to have values of ~100 m at most? Could you verify that those values are being plotted the way you expect? I also think it would be helpful if you could expand the Δdepth color scale so that Figure 5 is not entirely saturated – the point that it is uniformly positive could still be made while illustrating the magnitude of the deviation (which, right now, can only be inferred to be >= 320 m).

Figures 5 and 6 show the difference in isochrone depth below the surface while figure 7 shows the surfaces at an absolute elevation scale. Note that $BEST_{all}$ simulates a lower surface topography, which explains the mismatch. We added a short explanation to the end of the caption of figure 7.
With regard to the color scale in figure 5 and 6, it was carefully chosen as a) a compromise that works for both figures so that they can be compared, and b) so that the Δdepth scale is 1/10 of the total depth, using this relative mismatch as an intuitive reference. When finding a scale that works for both figures, a slight preference was given to figure 6 and the visibility of depth anomalies in $BEST_{all}$, because it is the more important simulation.

2. Figure 11 and line 332 – I understand the distinction you've made in the response to the previous review regarding "younger" vs "older" when describing bias, but I still stumble quite a bit with the language describing the orange curve in Figure 11. Here, you state there is an old bias (line 332). It would be helpful to be explicit about what specifically you are describing as biased (the Eulerian derived ages or the true model) and what they are biased relative to (the observed ages at those sites, the true ages of the model selected as optimal using the layer following scheme, or the true ages of the model selected based on the optimal Eulerian derived ages). By the

time I get to this sentence, I am already convinced by the previous paragraph that the Eulerian method calculates inaccurately old ages, so I don't think it adds value to say here that the Eulerian method is biased relative to the layer following method. Instead, the new piece of information is in the resulting ice sheet model bias. If you optimize your model to match the Eulerian ages to the observed ages, your model will actually produce an ice sheet that is younger than the true ice sheet (because the Eulerian ages that you've fit to are older than the true ages of the ice sheet, as shown by the layer following scheme). Phrasing things this way has the advantage that it is immediately apparent from Figure 11 -- the orange curve falls to the left of both the solid blue and the black curve.

Thank you very much for these comments. We followed the suggestions closely.

Beyond these two points of clarification, I only have a few line-item comments. I really enjoyed the paper!

Line-Item Corrections:
Page #: 2
Line #: 55
I'm not sure what you mean here by "spatial patterns that differ on the various isochronal surfaces." Patterns of what? Do you mean spatial patterns for isochrone depth error? It would help to be specific here.

This has been changed to "... spatially complex patterns of isochrone depth …".

Page #: 3
Line #: 72
It might be worth mentioning that N is the effective pressure assuming no support from a pressurized subglacial hydrologic system, just to be clear that effective pressure could have another term that is omitted here.

Done

Page #: 10
Line #: Figure 4
I think this is a great addition! I had one clarifying question -- the vertical interpolation you refer to here is for the Yelmo velocities, right? If so, should that either be in blue, or be the step following the input of the velocities from Yelmo? It might be helpful to state what is being interpolated and what is being advected in those two boxes, just to be totally explicit.

It is true that the previous version of the figure was not optimal. This has been changed now.

Page #: 11
Line #: 229
You're missing the section reference here -- I think it should read: "... by discussing two examples from the ensemble in section 3.1, followed by..."

Done

Page #: 12
Line #: Figure 5
As mentioned above, I think it would be helpful to expand the color scale here so that the interior values are not saturated (to get a sense for the full magnitude of the error).

Please see our answer above. We prefer to keep the figure unchanged.

Page #: 13
Line #: Figure 6
The panel showing error for the 11.7 ka error seems to have values with a maximum between 80 and 160 m in the Greenland interior, but that is quite different from the ~250 m errors shown in Figure 7. Is there something I'm missing here?

Please see our answer above. The figures are correct.

Page #: 17
Line #: 301
The word "perspective" here is vague -- I think it might be better to rephrase to something more specific, for example: "... the high resolution of our model in the temporal domain allows for a different and complementary analysis, using the full age profile at certain locations to constrain model performance."

Done

Page #: 17
Line #: 317
To assist the reader and maintain parallel structure here, it would be useful to point out that the Eulerian tracer is the dashed blue line in Fig. 11 and the isochronal tracing scheme is solid blue. The sentence as written references both methods but only one line in the figure, which can be confusing. IE: "Comparison between the Eulerian age tracer (dashed blue line) and the ioschronal age tracer (solid blue line) applied to the BEST_all simulation shows clear disagreement between the two methods for defining the depth-age scale."

Done

Page #: 17
Line #: 325-332
This is the section that I got a bit hung up on. I apologize if I said the opposite of this in my previous review (I think I might have...), but I realize now this sentence would be clearer if you talked about it in terms of parameter selection instead of model calibration. I tried my hand at rephrasing to clarify the points that confused me, with the hope that my attempted rephrasing will help you see where my issues were: "The spurious bias toward older ice with the Eulerian scheme has a noticeable effect on model behavior when parameter optimization is based on its output ages (as in the model plotted in orange, Fig. 11). Here we show the ages generated using the isochronal scheme, derived from model output from an ice sheet simulation using parameters chosen to optimize the quality of fit for isochrones generated using the Eulerian age tracer. Note that this is a different simulation from the one plotted in blue, which uses parameters chosen based on quality of fit for isochrones generated using the isochronal scheme."

Then, in line 332, I think the salient point is that models which optimize their boundary conditions using Eulerian derived ages will produce ice sheets with true ages that are younger than observations (as shown in Figure 11). This makes a clear distinction with the previous paragraph. There, you show that the Eulerian method produces older ages than the isochronal method -- here you show that model optimization based on Eulerian ages biases the ice sheet toward younger ice deeper in the column.

We followed these suggestions closely in our revised text.

---

## Author Response (AR3)

Thank you for your diligent work and encouragement. Please find our reply to your comments below.

**General Comments:**
P1 #l2: The IRH geometry is also imprinted by basal melting.

We rephrased this sentence.

P2 #l38: If I remember well Martin et al., 2009 is exclusively related to Antarctica. I suggest adding "…of the Greenlandic and Antarctic ice sheets"

It is true that the work of Martin et al. (2009) is primarily motivated by field sites in Antarctica. We now realize that this reference is not ideal, because it does not simulate the "three-dimensional stratigraphy". It has been removed from the revised text.

P3 Eq (1): Define \tau_b also in text.

Respectfully, \tau_b is defined as the basal friction by the text immediately preceding equation 1, in exactly the same way as \beta is defined by the text leading up to equation 2, E of equation 3, etc.

P3 L79/L95 Consider rephrasing: I don't agree that ice streams and ice shelves do not have a preferred crystal orientation fabric orientation. From what I have heard, e.g., from studies on the onset of NEGIS the contrary seems to be the case. Also measurements, e.g., on Whillans Ice Stream (Jordan et al., https://doi.org/10.1017/aog.2020.6) show ice anisotropy. I am not asking you to change anything on the modelling side, but the simplifications applied (ice anisotropy is "only" a function of vertical shearing) should be treated with more caution and can be explained in a larger context.

Thank you for pointing this out to us. We weakened the statements on lines 79 and 95.

P5 L113: "figure 1" -> "Figure 1"

Done

P5 L150: Is it really noise or climate-variability on timescales < 10kyr?

This refers to climate variability. The text has been revised accordingly.

Consider adding coordinates to Figs 2 and 3.

We did consider adding coordinates to all of our figures showing maps, but decided against it. All maps show stereographic projections, so that latitude lines are curves and meridians converge. This makes the figures very busy. At the same time, these coordinates do not help to locate specific regions more than guidance by the salient features of the Greenland coastline.

Section 2.4 Consider adding a sentence or two for what would be required if other users want to apply a different host model. How does the coupling look like in practice?

Moreover: Can this approach also accommodate inclusion of isochronal surface with unknown age?

A sentence has been added to make this more clear.

Section 3.3: The comparison with the Eulerian age trace is great, however, when reading through the paper I realized that no details were provided on the implementation of the Eulerian tracer. Specifics can be important here and I suggest to provide some numerical context, possibly by referencing the specific discretization scheme applied (cf. Greve et al., 2002, https://doi.org/10.3189/172756402781817112).

We added the short subsection 2.1.3 to provide some detail on the Eulerian scheme.

Consider acknowledging the input of the reviewers in the acknowledgements.

Of course!

I appreciate your supplements with the python function, however, when I tried using it I got the following error:
| => python read_data.py
Traceback (most recent call last):
File "read_data.py", line 9, in
ens = pickle.load(open('lhs_ensemble.pickle','rb'))
ModuleNotFoundError: No module named 'yelmo_utils'

We now supply the yelmo_utils module and a minimally updated version of the read_data.py script. The only difference in the latter is the publication date of the study.